# SepLLM: Accelerate Large Language Models by Compressing One Segment into One Separator

Guoxuan Chen [1 2]  Han Shi [1]  Jiawei Li [1]  Yihang Gao [2]  Xiaozhe Ren [1]  Yimeng Chen [3]  Xin Jiang [1]
Zhenguo Li [1]  Weiyang Liu [4]  Chao Huang [2]

## Abstract

Large Language Models (LLMs) have exhibited exceptional performance across a spectrum of natural language processing tasks. However, their substantial sizes pose considerable challenges, particularly in computational demands and inference speed, due to their quadratic complexity. In this work, we have identified a key pattern: certain seemingly meaningless separator tokens (*i.e.*, punctuations) contribute disproportionately to attention scores compared to semantically meaningful tokens. This observation suggests that information of the segments between these separator tokens can be effectively condensed into the separator tokens themselves without significant information loss. Guided by this insight, we introduce SepLLM, a plug-and-play framework that accelerates inference by compressing these segments and eliminating redundant tokens. Additionally, we implement efficient kernels for training acceleration. Experimental results across training-free, training-from-scratch, and post-training settings demonstrate SepLLM's effectiveness. Notably, using the Llama-3-8B backbone, SepLLM achieves over 50% reduction in KV cache on the GSM8K-CoT benchmark while maintaining comparable performance. Furthermore, in streaming settings, SepLLM effectively processes sequences of up to 4 million tokens or more while maintaining consistent language modeling capabilities.

## 1. Introduction

Transformer-based models (Vaswani et al., 2017) have exhibited exceptional performance across a wide range of

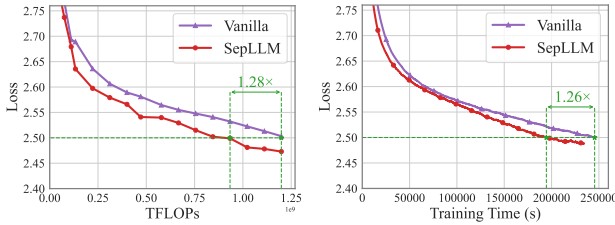

*Figure 1.* The loss comparison between vanilla Transformer and the proposed SepLLM. SepLLM achieves lower loss *w.r.t* different computation costs and different training time consistently.

tasks, including natural language processing (Zhang et al., 2020; Raffel et al., 2020), computer vision (Dosovitskiy et al., 2020), and scientific machine learning (Geneva & Zabaras, 2022). However, vanilla Transformers that rely on next-token prediction face significant computational challenges, particularly when scaling to larger models and longer contexts. These computational inefficiencies significantly impact both inference speed and training time.

The core challenge underlying these efficiency issues is the self-attention module, which exhibits quadratic complexity with respect to the number of input tokens. Research on efficient Transformers in LLMs primarily follows two major directions. The first approach focuses on linear attention (Katharopoulos et al., 2020; Schlag et al., 2021), replacing the vanilla self-attention module with alternatives that achieve linear complexity. However, these modifications make the architecture significantly different from traditional self-attention, preventing direct utilization of powerful pretrained Transformer models. The second approach emphasizes KV cache optimization (Xiao et al., 2024a; Zhu et al., 2024; Xiao et al., 2024b; Li et al., 2024), aiming to eliminate redundant KV cache to accommodate longer input contexts. For example, Xiao et al. (2024a) introduced an adaptive mechanism that selectively retains essential tokens and their KV based on cumulative attention scores. Similarly, Zhu et al. (2024) proposed a token selection strategy with controlled sparsity, achieving near-lossless acceleration. While promising, these training-free methods adapt poorly to the training stage, resulting in discrepancies between training and inference performance. StreamingLLM (Xiao et al., 2024b) represents a notable attempt to address these lim-

[1]Huawei Noah's Ark Lab [2]The University of Hong Kong [3]Center of Excellence for Generative AI, KAUST [4]Max Planck Institute for Intelligent Systems, Tübingen. Correspondence to: Han Shi <shi.han@huawei.com>.

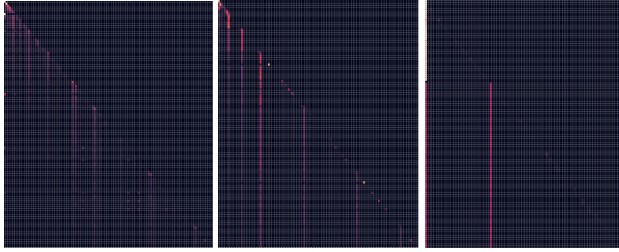

*Figure 2.* The visualization for attention scores of different layers given the input "Natalia sold clips to 48 of her friends in April, and then she sold half as many clips in May. ...". Note that the separator tokens like "," and "." contribute massive attentions.

itations by preserving attention sinks and local tokens to reduce computation and memory overhead. However, it omits many intermediate tokens, resulting in performance degradation compared to standard Transformers.

To gain better understanding of the intrinsic mechanism of LLMs, we analyze attention patterns across different samples. Figure 2 illustrates the attention distribution when Llama-3-8B-instruct (Dubey et al., 2024) processes a math problem (complete results are presented in Appendix A). Surprisingly, rather than focusing on semantically meaningful tokens (such as nouns and verbs), LLMs tend to prioritize attention to seemingly "meaningless" separator tokens (like "." or "\n") for information retrieval. This observation suggests that segment information is compressed and embedded into these separator tokens, enabling efficient information retrieval without direct extraction from content tokens.

Inspired by this observation, we introduce SepLLM, a new language modeling perspective as well as an efficient transformer architecture featuring a data-dependent sparse attention mechanism that selectively retains only initial, neighboring, and separator tokens while dropping other tokens. The training-free SepLLM performs comparably to vanilla Transformer, which validates our hypothesis that segment information is effectively compressed into separator tokens. More importantly, we integrate SepLLM into the training stage (including both training from scratch or finetuning) and implement a hardware-efficient kernel based on Flex-Attention (PyTorch, 2024). This integration reduces the discrepancies between training and inference that are present in previous approaches. As demonstrated in Figure 1, SepLLM consistently achieves lower loss compared to vanilla Transformer given the same computational costs or training time. Moreover, SepLLM reduces computational costs by 28% and training time by 26% while achieving the same training loss. Our contributions are summarized as follows:

- We analyze attention patterns by visualizing token-level attention scores, revealing that initial, neighboring, and separator tokens consistently receive high attention weights. Such empirical findings motivate us to propose SepLLM,

**a new language modeling perspective** and a simple yet effective framework to accelerate inference.

- Through targeted masking experiments on well-trained LLMs, we demonstrate that separator tokens contain crucial information and are essential for model performance. This finding suggests that sequences are initially segmented by separators, with segment information being compressed into these frequently-attended separator tokens while redundant specific tokens can be discarded.

- We conduct comprehensive experiments to validate SepLLM's effectiveness across various tasks, datasets, and backbone models, examining performance in training-free, training-from-scratch, and post-training settings.

- We have made our implementation publicly available at **sepllm.github.io**. Our codebase supports efficient multi-node distributed training with accelerated attention module *Sep-Attention* and also supports numerous existing Fusion Operators to accelerate the training process, such as *fused rope* (Su et al., 2024), *fused layer norm*, etc.

## 2. Related Work

**KV Cache Compression**. Recent research has focused on overcoming LLMs' limitations in processing extensive contextual inputs. FastGen (Ge et al., 2024) proposes an adaptive KV cache management method, optimizing memory usage by customizing retention strategies for different attention heads. SnapKV (Li et al., 2024) enhances efficiency through KV cache compression, utilizing attention scores to select and cluster significant positions. $H_2O$ (Zhang et al., 2023) implements a dynamic token retention policy, balancing recent and historically important information to optimize memory use. StreamingLLM (Xiao et al., 2024b) expands LLMs' capabilities to handle infinite sequence lengths without fine-tuning, by reserving attention sinks and local tokens. QuickLLaMA (Li et al., 2025) proposes to evict the query-aware KV cache for inference acceleration. PyramidInfer (Yang et al., 2024) and PyramidKV (Zhang et al., 2024) modify the KV cache capacity across different layers, prioritizing larger allocations in the lower layers while reducing those in the upper layers. However, most works in this category cannot be applied into training phase.

**Sparse Attention**. Sparse attention involves creating sparse attention matrices by limiting attention to predefined patterns, such as local windows or fixed-stride block patterns. Beltagy et al. (2020) combine dilated local window attention with task-specific global attention. BigBird (Zaheer et al., 2020) proposes a linear-complexity attention alternative using global tokens, local sliding-window attention, and random attention. In comparison, SparseBERT (Shi et al., 2021) proposes a differentiable attention mask algorithm to learn the attention mask in an end-to-end manner. Note that most works about sparse attention are using fixed masks and

built on BERT (Devlin et al., 2019) families. In comparison, our proposed SepLLM is mainly built on GPT (Brown et al., 2020) series and its attention masks are data-dependent.

## 3. Method

### 3.1. Fundamental Design

From Figure 2, we can observe that within a given input context, seemingly "meaningless" separator tokens receive higher attention scores compared to tokens with actual semantic meanings. Therefore, we propose a novel Transformer architecture where, for a certain layer of the Transformer (*i.e.*, a self-attention layer), each token in the input can only see a portion (not all) of the hidden states of tokens preceding the current token, outputted by the previous transformer layer. This subset of tokens includes a number of initial words (*e.g.*, attention sinks (Xiao et al., 2024b)), all the separator tokens before the current token, and the closest *n* tokens to the current token. Details are as follows.

**Initial Tokens**. When using the sliding window mechanism (Beltagy et al., 2020) for generation, removing the key-value (KV) pairs corresponding to the initial tokens in the KV cache results in a noticeable increase in the perplexity of generated tokens, a phenomenon mentioned by Xiao et al. (2024b). The initial few tokens are also referred to as attention sinks. We retain this setup and further validate the role of initial tokens in subsequent experiments. Usually, *a* initial tokens are kept.

**Separator Tokens**. From Figure 2, we can observe that within a given input context, seemingly "meaningless" separator tokens (such as commas, periods, exclamation marks, semicolons, etc.) that segment sequences receive higher attention scores compared to semantically meaningful tokens (such as nouns or verbs). Therefore, we hypothesize that these separators may compress the information of the text segments naturally segmented by them, such that when the Transformer generates new tokens, it only needs to reference the information contained in these separators to extract the information pertaining to those text segments. Hence, in a training-free scenario, we employed this strategy and achieved similar results to the original model based on full attention across many tasks. Furthermore, to reinforce the effect of using separators to compress information within their respective segments, we employed training-from-scratch and post-training approaches to compel the model during training to restrict the current token from accessing all information from distant preceding text, *i.e.*, in each segment, only the separator representing its segment is visible to the current token (with other tokens being masked, see Figure 3). After training in this manner, the information within segments is forced to be condensed into the separators, leading the Transformer's probability distribution for predicting the next word closely resembling that of the original Trans-

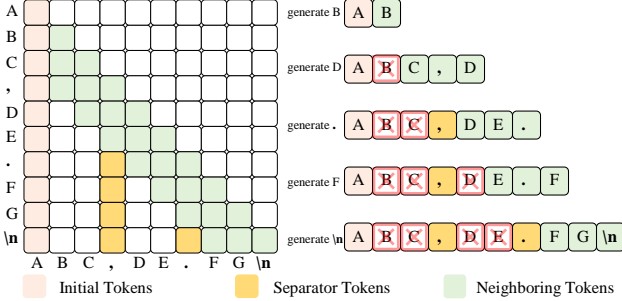

*Figure 3.* The overall paradigm of SepLLM. The left side illustrates the attention mask in the training or pre-filling stage given the input "ABC,DE.FG\n". The right side illustrates the KV cache management in the generation stage.

former with full attention. See more in Appendices H and I.

**Neighboring Tokens**. Language tasks usually exhibit strong local dependencies and interactions, since adjacent tokens often form coherent phrases or have dependencies that are required to be captured. Neighboring tokens usually help form locally smooth and coherent contexts, allowing the model to generate sentences that are reasonable within the immediate context. The neighboring tokens, also referred to as local attention or sliding-window attention, are considered in various efficient Transformers (Xiao et al., 2024b; Zhang et al., 2023) and we have also adopted this approach, with the number of preceding tokens closest to the current token denoted as "*n*".

### 3.2. Overall Pipeline

We split the overall pipeline of our proposed SepLLM into training/pre-filling stage and generating stage. We also provide a theoretical analysis about the *Universal Approximation* of SepLLM in Appendices J and K.

**Training/Pre-filling.** During the training/pre-filling stage of SepLLM architecture, we do not need to multiply all query vectors corresponding to tokens in the input context with all the key vectors. It is sufficient to just multiply the vectors of the query-key pairs corresponding to the highlighted elements in the mask matrix shown in Figure 3. The formulation can be illustrated in the following.

$$\mathbf{A} = \text{Softmax}\,(\Lambda)\,, \Lambda = \frac{\text{Mul}\left(\mathbf{Q},\ \mathbf{K}^{\top}\,|\,\mathbf{M}\right)}{\sqrt{d_k}}$$

$$\mathbf{O} = \mathbf{A} \cdot \mathbf{V} \qquad (1)$$

where $\mathbf{Q} \in \mathbb{R}^{m \times d_k}, \mathbf{K} \in \mathbb{R}^{m \times d_k}$ are the matrices of query and key for one attention layer, in which each row vector $\mathbf{Q}_i, \mathbf{K}_j$ correspond to the query of $i$-th token and the key of $j$-th token in the input context with sequence length $m$. $d_k$ denotes the dimension for key and query vectors.

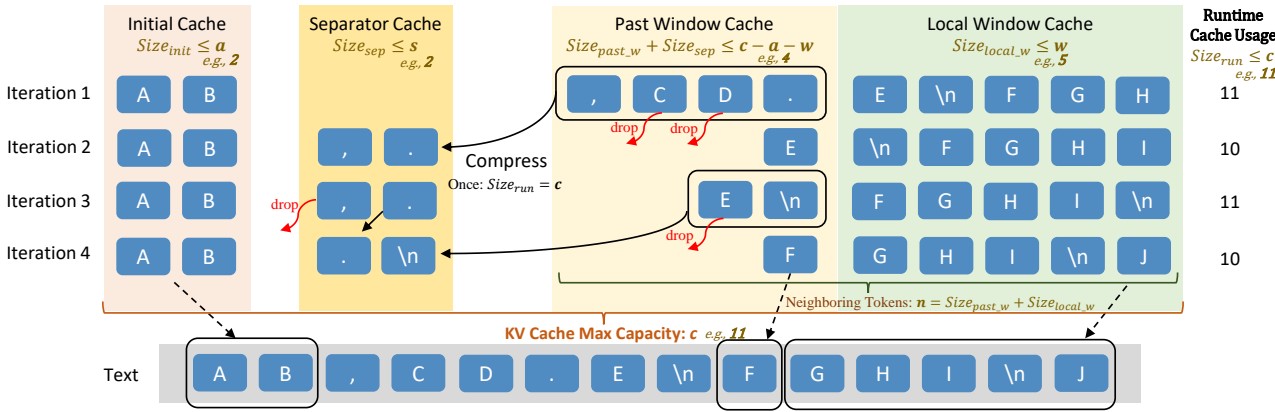

*Figure 4.* Overall framework of the proposed SepLLM tailored for streaming applications. The KV pairs are stored in four cache blocks (displayed as four columns), and are updated in each iteration (shown in a single row). Once the runtime usage $Size_{run}$ reach the max capacity **c**, SepLLM move KV caches of separator tokens in Past Window Cache into Separator Cache and drop other KV caches.

$\Lambda, \mathbf{A} \in \mathbb{R}^{m \times m}$ are the raw and final attention maps, respectively. $\mathbf{V} \in \mathbb{R}^{m \times d_v}$ is value matrix of dimension $d_v$ and $\mathbf{O} \in \mathbb{R}^{m \times d_v}$ denotes the output for the current attention layer. $\mathrm{Mul}(\cdot)$ represents a sparse matrix multiplication function which can be optimized by methods like Zhu et al. (2024) and we also implement our own module named *Sep-Attention* to accelerate this process. $\mathbf{M} \in \mathbb{B}^{m \times m}$ is a binary mask matrix[1] used as a parameter for $\mathrm{Mul}(\cdot)$:

$$\Lambda_{i,j} = \begin{cases} \mathbf{Q}_i^\top \mathbf{K}_j / \sqrt{d_k}, & \text{if } \mathbf{M}_{i,j} = 1 \\ -\infty, & \text{if } \mathbf{M}_{i,j} = 0 \end{cases}. \quad (2)$$

where $\Lambda_{i,j}, \mathbf{A}_{i,j}, \mathbf{M}_{i,j}$ are the elements in the $i$-th row and $j$-th column of matrices $\Lambda, \mathbf{A}, \mathbf{M}$, respectively. Since $\mathbf{A}_{i,j} = 0$ if $\Lambda_{i,j} = -\infty$, the tokens that are not *Initial*, *Separator*, and *Neighboring* tokens will be masked by $\mathbf{A} \cdot \mathbf{V}$ in Equation 1. This strategy (Equation 1) applies to all heads of multi-head attention (Vaswani et al., 2017).

**Generation.** The management of the KV cache during the generation stage for this *Fundamental Design* (Section 3.1) is also intuitive. As shown in the right side of Figure 3, when generating a new token, we only preserve the KV cache for the *Initial*, *Separator*, and *Neighboring* tokens. Therefore, the KV cache in the proposed SepLLM is much smaller and requires less memory. Ideally, based on SepLLM, the perplexity of generating the next word is comparable to that of the original Transformer with full attention.

### 3.3. Tailored Streaming Design

In real-world scenarios, there are numerous streaming applications such as multi-round dialogues, where long interactions are expected (Xiao et al., 2024b). Hence, we expect SepLLM to handle infinite input without significantly sacrificing efficiency and performance, especially for streaming

applications. As discussed in *Fundamental design* (Section 3.1), SepLLM can save a substantial amount of KV cache by retaining only the KV for separator, neighboring, and initial tokens. However, as the number of input tokens increases, the number of separators in KV cache will also accumulate endlessly, which is not feasible for streaming settings. Therefore, we propose *Tailored Streaming Design* for streaming scenarios.

**Framework.** Figure 4 illustrates the SepLLM's processing architecture for streaming applications. The diagram depicts multiple iterations, with each row representing a distinct processing step. The system simultaneously maintains four specialized cache blocks: Initial Cache, Separator Cache, Past Window Cache, and Local Window Cache. Specifically, Initial Cache captures the attention sinks proposed by Xiao et al. (2024b). Local Window and Past Window Caches store the KV for consecutive tokens, with Past Window Cache serving as an overflow buffer for the Local Window Cache. Separator Cache retains the KV for separators which contain condensed segment information.

To describe the cache management strategies, we denote the runtime usage of the four caches as $Size_{\text{init}}$, $Size_{\text{sep}}$, $Size_{\text{past\_w}}$, and $Size_{\text{local\_w}}$, respectively. The runtime usage across all KV caches is defined as $Size_{\text{run}} := Size_{\text{init}} + Size_{\text{sep}} + Size_{\text{past\_w}} + Size_{\text{local\_w}}$, which satisfies $Size_{\text{run}} \leq$ **c**. The number of continuous Neighboring tokens is defined as $\boldsymbol{n} := Size_{\text{past\_w}} + Size_{\text{local\_w}}$. Notably, $\boldsymbol{n}$ is a function of the input sequence length $m$ (together with the specific input dataset $\mathcal{D}$) rather than a fixed hyperparameter for streaming setting. For clarity, we detail the preset hyperparameters of this caching system as follows (*Note*: $\boldsymbol{a} + \boldsymbol{s} + \boldsymbol{w} < \boldsymbol{c}$).

- **c**: The maximum capacity of the entire KV cache.
- **a**: The maximum capacity of Initial Cache.
- **s**: The maximum capacity of Separator Cache.

---

[1] $\mathbb{B} := \{0, 1\}$, which is a binary set.

- **$w$**: The maximum capacity of Local Window Cache. Notably, $w$ is also the minimum value of $n$ after runtime KV cache usage $Size_{\text{run}}$ reaches $c$ for the first time.

During streaming sequence generation, SepLLM will firstly fill Initial Cache and then Local Window Cache. After $Size_{\text{local\_w}}$ reaches $w$, subsequent tokens are directed to Past Window Cache. Compression is triggered when $Size_{\text{run}}$ reaches $c$ (iteration 1 in Figure 4), where separator tokens in Past Window Cache are moved to Separator Cache and other tokens are discarded. When Separator Cache reaches its capacity $s$ at some total input length $m_0$, $n$ enters a periodic pattern. Specifically, for $m > m_0$, $n$ follows a periodically linear function bounded by $w$ and $c - a - s$. The detailed evolution of KV caches is illustrated in Appendix B. To analyze the average usage of KV cache, we define $\bar{n}_m := \frac{1}{m} \sum_{k=1}^{m} n(k)$. According to the linearity and periodicity, we have

$$\lim_{m \to \infty} \bar{n}_m = \frac{w + c - a - s}{2}. \tag{3}$$

For the average runtime KV cache usage of infinite-length sequence generation, we have

$$\lim_{m \to \infty} \overline{Size_{\text{run}}} = \lim_{m \to \infty} \bar{n}_m + a + s$$
$$= \frac{w + c + a + s}{2} < c. \tag{4}$$

**Positional Encoding.** Our positional encoding strategy for streaming settings is the same as the state-of-the-art StreamingLLM (Xiao et al., 2024b), designed specifically for infinite-length inputs, where we focus on positions within the cache instead of those in the original text.

# 4. Experiments and Results

## 4.1. Experimental Settings

We evaluate our proposed SepLLM on the following tasks, *i.e.*, training-free, training-from-scratch, post-training, and streaming applications.

**Model.** Two popular model families, *i.e.*, Pythia (Biderman et al., 2023) and Llama-3 (Dubey et al., 2024), are employed for evaluation. Specifically, Pythia-160m-deduped is used as the backbone in the training-from-scratch tasks since the model, data, configurations, and checkpoints are all open-source and the training results are reproducible. As for post-training settings, we take Pythia-1.4B-deduped as our backbone model. Even though Llama-3 exhibits powerful performance on various downstream tasks, the training experimental details are not available. Therefore, we only use Llama-3 for training-free and streaming tasks.

**Training Datasets.** In the training-from-scratch and post-training tasks, the deduplicated Pile (Gao et al., 2020) is utilized for training, which contains about 207B tokens. And

all other configurations are the same as the corresponding settings as Pythia (Biderman et al., 2023). Specifically, the training epoch is set to 1.5 epoch (143000 steps with the global batch size as 1024), which means about 300B tokens in total are utilized for training from scratch, which is identical to Pythia (Biderman et al., 2023).

**Parameter Setting.** The official 93,000-step checkpoint of Pythia-1.4B-deduped model is used to conduct post-training, which corresponds to just completing one epoch of training on the deduped Pile dataset (Gao et al., 2020). And [".", ",", "?", "!", ";", ":", " ", "\t", "\n"] are separator tokens used for all evaluations. More specific experimental settings are introduced in the respective experiment sections.

## 4.2. Training-Free

We evaluate the proposed SepLLM architecture in the training-free tasks based on the popular Llama-3-8B-Instruct model (Dubey et al., 2024).

**Benchmarks.** The representative and commonly-used GSM8K-CoT (Cobbe et al., 2021) and MMLU (Hendrycks et al., 2021)) are adopted. GSM8K-CoT (Cobbe et al., 2021) tests a model's ability to solve mathematical problems by evaluating its reasoning and step-by-step problem-solving skills. CoT (Wei et al., 2022) means the ability to simulate a reasoning process by breaking down complex problems into a series of logical steps. And the default 8 shots are adopted. MMLU (Hendrycks et al., 2021) assesses a model's general knowledge and reasoning ability across a wide range of subjects, such as history, science, mathematics and so on. The commonly-used 5-shot setting is used for MMLU.

**Results.** The experimental results for training-free are shown in Table 1. "Vanilla" represents the original Llama-3 model with full attention, while "StrmLLM" represents StreamingLLM (Xiao et al., 2024b). $n$ means the number of KV for Neighboring Tokens we retain. For SepLLM, all the KV for Separator Tokens are kept and for the setting *SepLLM (n=256)*, we find that SepLLM exhibits comparable performance in both multi-step mathematical CoT task and multidisciplinary knowledge reasoning tasks, when compared to the full-attention Llama-3. SepLLM achieves this using only 47.36% of the KV utilized by the original Llama-3 for reasoning, indicating SepLLM's capability of modeling both the contexts requiring multi-step logical analysis and those involving multi-domain knowledge reasoning while retaining only 50% original KV.

*StrmLLM (n=256)* setting corresponds to removing all separators' KV from *SepLLM (n=256)* setting, except for those in Neighboring and Initial tokens. We observe a noticeable decrease in both mathematical analysis and multidisciplinary knowledge reasoning abilities for *StrmLLM (n=256)*.

| | GSM8K-CoT | | | MMLU | | | | | |
|---|---|---|---|---|---|---|---|---|---|
| | flexible | strict | r.KV (%) | humanities | stem | social | other | Overall | r.KV (%) |
| Vanilla | 77.79 | 77.26 | 100.00 | 60.49 | 56.61 | 76.50 | 72.19 | 65.72 | 100.00 |
| StrmLLM (*n*=380) | 70.89 | 71.42 | 47.54 | 57.73 | 54.46 | 74.39 | 70.13 | 63.39 | 52.50 |
| StrmLLM (*n*=256) | 69.67 | 68.61 | 26.00 | 62.10 | 54.49 | 73.06 | 69.78 | 62.10 | 37.73 |
| SepLLM (*n*=256) | 77.18 | 77.18 | 47.36 | 57.66 | 56.49 | 76.21 | 72.19 | 64.68 | 44.61 |

*Table 1.* Evaluation results and average *runtime* KV cache usage for training-free experiments on GSM8K-CoT 8-shots and MMLU 5-shots. For SepLLM and StreamingLLM, three initial tokens' KV are kept for this experiment. *r*.KV (%) here represents the ratio of KV usage at *runtime* for the respective method compared to Vanilla. See more results in Appendices I and Table 17.

| Method | ARC-c | ARC-e | LBD-ppl | LBD-acc | LogiQA | PIQA | SciQ | Atten. (%) | r.KV (%) |
|---|---|---|---|---|---|---|---|---|---|
| Vanilla | 20.14 | 46.80 | 34.83 | 33.28 | 23.81 | 62.84 | 81.50 | 100.00 | 100.00 |
| StrmLLM(*n*=64) | 20.65 | 47.39 | 44.03 | 26.74 | 21.97 | 63.82 | 75.80 | 16.58 | 15.28 |
| SepLLM(*n*=64) | 19.62 | 46.46 | 40.08 | 28.97 | 26.42 | 63.82 | 80.10 | 25.83 | 25.40 |
| SepLLM(*n*=128) | 19.97 | 47.35 | 30.16 | 33.18 | 22.73 | 64.64 | 82.60 | 35.64 | 32.27 |
| SepLLM(*n*=64,H) | 20.73 | 48.44 | 36.54 | 30.45 | 25.35 | 64.36 | 80.60 | 32.01 | 31.58 |
| SepLLM(*n*=64,H/T) | 21.42 | 47.26 | 33.41 | 32.80 | 22.73 | 63.98 | 81.20 | 38.18 | 37.75 |

*Table 2.* The performance of downstream tasks and the average *runtime* KV cache usage in the training-from-scratch setting.

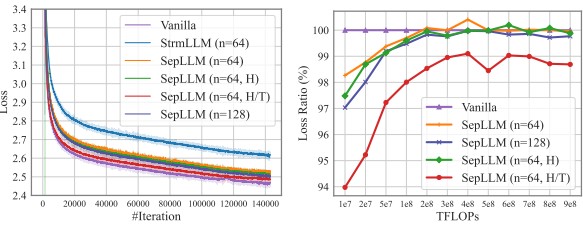

(a) Loss *w.r.t* steps      (b) Loss Ratio *w.r.t* FLOPs

*Figure 5.* Training loss curves for training from scratch. 5(b) shows the ratios of the loss values of different methods to that of Vanilla with respect to FLOPs.

*StrmLLM (n=256)* utilizes only 26.00% and 37.73% of the KV for the GSM8K and MMLU tasks, respectively, which are less than *SepLLM (n=256)* (47.36% and 44.61% respectively). Consequently, we increase the *n* of *StrmLLM* to 380, aligning the kept KV on GSM8K to be equal to *SepLLM (n=256)* (approximately 47%, while on MMLU task, *StrmLLM (n=380)* retains 52.5% of the KV, significantly higher than *SepLLM (n=256)*). This leads to improved performance compared to *StrmLLM (n=256)*. However, it still remains lower than the full-attention Llama-3 and *SepLLM (n=256)*. This indicates that the KV of separators indeed encapsulates information contained within their respective segments, and removing them significantly impacts the Transformer's understanding and reasoning abilities.

### 4.3. Training from Scratch

We train the original Pythia-160m-deduped model as well as the Pythia-160m-deduped model modified with the SepLLM (and StreamingLLM) architecture on the Pile dataset for 143,000 steps using a global batch size of 1024 (involving approximately 300B tokens in total for training). All training configurations are consistent with Pythia (Biderman

et al., 2023) (see Section 4.1). And following Pythia (Biderman et al., 2023), we conduct tests on the following downstream tasks: ARC-Challenge and ARC-Easy (Clark et al., 2018), LAMBADA (Paperno et al., 2016) (for Perplexity and Accuracy), LogiQA (Liu et al., 2021), PIQA (Bisk et al., 2020), SciQA (Welbl et al., 2017). From the loss curves depicted in Figure 5 and the downstream performance in Table 2, we draw the following analysis.

**Neighboring Token Benefits.** Based on the experiments with the settings *SepLLM (n=64)* and *SepLLM (n=128)*, we find that during training, increasing Neighboring Tokens (*n*) leads to a faster decrease in the training loss curve (Figure 5). Furthermore, models trained with larger *n* exhibit stronger performance in downstream tasks (Table 2). This highlights the important role of neighboring tokens in contextual language modeling and downstream task inference.

**Hybrid Layer Benefits.** We find that employing a certain hybrid architecture is beneficial to both the training loss and the performance on downstream tasks. For instance, by modifying only the first self-attention layer to full attention in the experiment corresponding to *SepLLM (n=64)* (denoted as *SepLLM (n=64,H)*), there is a moderate optimization in both the training process and downstream tasks. If both the first and last attention layers are changed to full attention (denoted as *SepLLM (n=64,H/T)*), this optimization becomes more pronounced. For example, LAMBADA perplexity decreases from 40.08 for *SepLLM (n=64)* to 36.54 for *SepLLM (n=64,H)* and 33.41 for *SepLLM (n=64,H/T)*.

**Separators' Role.** The experiment with the setting *StrmLLM (n=64)* corresponds to *SepLLM (n=64)*, but does not consider separators other than those in Neighboring and Ini-

| Arch. | StrmLLM | SepLLM | | | | Vanilla |
|---|---|---|---|---|---|---|
| Setting | $n$=64 | $n$=64 | $n$=128 | $n$=64,H | $n$=64,H/T | full |
| FLOPs (%) | 70.11 | 71.77 | 72.58 | 72.83 | 73.90 | 100.0 |
| Atten. (%) | 6.43 | 17.21 | 22.48 | 24.11 | 31.01 | 100.0 |

*Table 3.* The comparison of FLOPs and Attention Map Ratios.

tial tokens. We observe a significant slowdown in the training loss decrease for *StrmLLM (n=64)*, and its performance deteriorates across various downstream tasks. This indicates that the KV corresponding to separators indeed contain information about the segments they belong to, which are beneficial to predicting subsequent tokens.

We also investigate the FLOPs and Attention Map Ratio (indicating the proportion of '1's in the lower triangle of the attention mask) required by the different architectures when trained on the same input data. As shown in Table 3, We find that SepLLM can significantly reduce FLOPs by approximately 30%. After plotting the loss ratios between SepLLM and Vanilla under the same FLOPs (see Figure 5(b)), we observe that SepLLM has lower loss than Vanilla. This indicates that our SepLLM architecture at least has a comparable ability to extract useful information from the dataset during training as Vanilla. Besides, the detailed wall-clock time per iteration and the wall-clock time speedups are illustrated in Appendix C and Figure 1.

### 4.4. Post-Training

Since training from scratch is time-consuming, we also conduct post-training experiments using 93000-step Pythia-1.4B-deduped checkpoint officially released by Pythia (see Section 4.1 for details). Figure 6 displays the loss curves for post-training, where *SepLLM (n=64, larger lr)* denotes we employ an entire cosine learning rate scheduler (including a warm-up process starting from 0) identical to that of original Pythia-1.4B-deduped from step 0. *SepLLM (n=64)* and *SepLLM (n=128)* utilize a cosine learning rate scheduler that continues to decay from the 93000th step. From Figure 6, it is evident that increasing *n* and appropriately raising the learning rate both facilitate the decrease in loss. Moreover, this also illustrates that SepLLM can achieve a swift transformation from a full-attention LLM checkpoint to a model that aligns with the requirements of the SepLLM architecture's embedding distribution through post-training.

### 4.5. Streaming Applications

SepLLM can also adapt well to streaming applications, where infinite-length interactions may occur. Here, we follow StreamingLLM (Xiao et al., 2024b) to validate the scenarios of infinite-length interactions using our *Tailored Streaming Design* on the commonly used PG19 dataset (Rae et al., 2020), which comprises 100 extensive literary works.

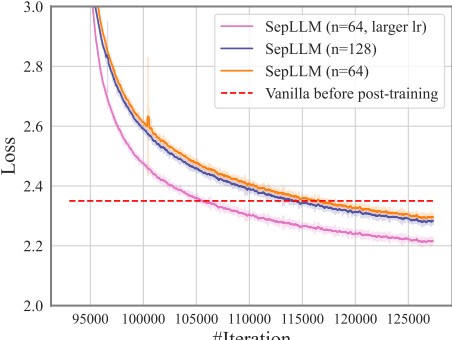

*Figure 6.* Training loss curves for the post-training setting.

| PG19 | 1M | 1.5M | 2M | 2.5M | 3M | 3.5M | 4M |
|---|---|---|---|---|---|---|---|
| StrmLLM | 39.5 | 38.2 | 38.3 | 37.6 | 36.4 | 35.8 | 36.1 |
| SepLLM ($s$=32) | 37.7 | 36.6 | 36.6 | 36.0 | 34.9 | 34.2 | 34.5 |
| SepLLM ($s$=64) | 37.1 | 36.0 | 36.1 | 35.4 | 34.3 | 33.7 | 33.9 |

*Table 4.* The perplexity comparison on the PG19 test set (Rae et al., 2020). For fair evaluation, we keep the entire KV cache capacity $c$ as 324 and Initial Cache capacity $a$ as 4 for both StreamingLLM and SepLLM. $w$=224, $s$=32/64 for SepLLM.

The results are shown in Table 4. We can observe that for the same KV cache capacity $c$, the average perplexity of predicting the next token through SepLLM remains consistently lower than that of streamingLLM (Xiao et al., 2024b) within the range from 1M to 4M input length. This once again verifies the ability of KV corresponding to separators to compress segment information and their impact on predicting the probability distribution of the next token.

We also test the end-to-end inference time of Vanilla, StreamingLLM and our SepLLM on PG19 (Rae et al., 2020) test set based on LlaMA-3-8B (Dubey et al., 2024). Based on the aforementioned settings, we used these LLMs to generate 20K and 64K tokens to evaluate their total inference time (wall-clock time), average perplexity, and average runtime KV cache usage. For both SepLLM and StreamingLLM, the maximum whole KV cache capacity was set to 800 (*i.e.*, $c$=800), and the Initial Cache capacity was set to 4 (*i.e.*, $a$=4). For SepLLM, we additionally set $s$=64 and $w$=256. The results are shown in Table 5.

Those results demonstrate that our SepLLM can achieve lower perplexity with less wall-clock time as well as lower average runtime KV usage, especially for longer sequences, given the same max KV cache capacity $c$.

### 4.6. Ablation Study

**Hyperparameters.** We conduct various ablation experiments specifically for long-input applications. This includes a detailed study of the impact of various hyperparameters across different text lengths (5K to 20K). The experimental results about $s$ and ($w$,$c$) pair are illustrated in Table 6 and Table 7 respectively. The conclusions are as follows.

| Length | Methods | $c$ | $r$.KV | ppl | time (s) |
|---|---|---|---|---|---|
| | Vanilla | 20K | 10K | 302.6 | 523.8 |
| 20K | StrmLLM | 800 | 800 | 31.5 | 341.2 |
| | SepLLM | 800 | 562 | 28.3 | 325.8 |
| | Vanilla | 64K | 32K | 1090.8 | 3380.6 |
| 64K | StrmLLM | 800 | 800 | 37.9 | 1096.0 |
| | SepLLM | 800 | 562 | 33.4 | 1049.7 |

*Table 5.* The average perplexity and running time comparison on the PG19 test set (Rae et al., 2020). $r$.KV means the average *runtime* KV cache usage in the generation process.

| $s$ | 5K | 10K | 15K | 20K | $r$.KV |
|---|---|---|---|---|---|
| 32 | 13.11 | 11.31 | 8.74 | 8.79 | 292 |
| 48 | 13.03 | 11.26 | 8.70 | 8.76 | 300 |
| 64 | 13.01 | 11.17 | 8.67 | 8.72 | 308 |

*Table 6.* The perplexity and average *runtime* KV cache usage of SepLLM with respect to different Separator Cache capacities ($s$) on WikiText (Merity et al., 2017), in which $a$=4, $w$=224, $c$=324.

- **$s$**: From Table 6, the capacity of Separator Cache affects the perplexity of long-text inference, as we find that increasing $s$ leads to a certain degree of perplexity reduction.

- **$c$ and $w$**: As can be seen in Table 7, $c$ and $w$ can impact the average perplexity in the scenario of long streaming input with lengthy text. Moreover, as they increase, the perplexity decreases accordingly.

**Settings.** We also perform the following experiments to validate the effectiveness of Initial Tokens and Positional Encoding's shifting for both StreamingLLM and SepLLM. The experimental results are shown in Table 8 and the discussions are as follows.

- **Initial Tokens**: Initial Tokens are crucial for modeling context of long streaming inputs, whether for SepLLM or StreamingLLM. Removing them has a significant impact on the perplexity of long texts. This conclusion is consistent with the paper (Xiao et al., 2024b).

- **Positional Encoding's Shifting**. Following streamingLLM, we conduct Positional Shifting for streaming applications, *i.e.*, we focus on positions within the cache rather than those in the original text. Table 8 shows that this shifting plays a crucial role, as removing it significantly increases the perplexity (StreamingLLM increases from around 13 to over 400). It is noteworthy that SepLLM, even without employing this shifting, only sees a perplexity increase to around 200, which is much lower than StreamingLLM. This further underscores the separators' role for the stability in predicting tokens.

**Separator Choices.** A series of ablation studies are also conducted on the choice of separators. For SepLLM, whether it is the *Fundamental Design* or the *Tailored Streaming Design*, the choice of separator types serves as a kind

| Method | $w$ | $c$ | $r$.KV | 5K | 10K | 15K | 20K |
|---|---|---|---|---|---|---|---|
| | 320 | 324 | 324 | 13.18 | 11.51 | 8.85 | 8.91 |
| StrmLLM | 512 | 516 | 516 | 12.87 | 11.37 | 8.74 | 8.78 |
| | 796 | 800 | 800 | 11.96 | 11.01 | 8.67 | 8.72 |
| | 224 | 324 | 308 | 13.01 | 11.17 | 8.67 | 8.72 |
| SepLLM | 320 | 516 | 452 | 12.91 | 11.26 | 8.67 | 8.72 |
| | 512 | 800 | 690 | 12.09 | 11.03 | 8.56 | 8.62 |

*Table 7.* Average downstream performance (ppl) over different input lengths and average *runtime* KV usage with different $c$,$w$ on WikiText, in which $a$=4 for both methods and $s$=64 for SepLLM.

| Method | initial | shift | 5K | 10K | 15K | 20K | $r$.KV |
|---|---|---|---|---|---|---|---|
| StrmLLM | ✓ | ✓ | 13.2 | 11.5 | 8.9 | 8.9 | 324 |
| StrmLLM | ✗ | ✓ | 14.6 | 13.2 | 10.8 | 10.9 | 324 |
| StrmLLM | ✓ | ✗ | 425.5 | 513.1 | 509.5 | 506.8 | 324 |
| StrmLLM | ✗ | ✗ | 409.4 | 540.5 | 527.5 | 558.2 | 324 |
| SepLLM | ✓ | ✓ | 13.1 | 11.3 | 8.7 | 8.8 | 292 |
| SepLLM | ✗ | ✓ | 14.9 | 14.3 | 12.4 | 12.5 | 290 |
| SepLLM | ✓ | ✗ | 192.7 | 214.6 | 175.0 | 174.4 | 292 |
| SepLLM | ✗ | ✗ | 226.4 | 264.7 | 227.5 | 228.8 | 290 |

*Table 8.* The perplexity and average *runtime* KV cache usage of SepLLM and StreamingLLM tested on WikiText (Merity et al., 2017). $c$=324, $a$=0/4 for both methods. $s$=32,$w$=224 for SepLLM.

of hyperparameter. Given that the types of common separators are limited (*e.g.*, the ones we adopt here [".", ",", "?", "!", ";", ":", " ", "\t", "\n"]), we recommend including all commonly used ones. A detailed discussion of the ablation studies on separators is provided in Appendix G.

### 4.7. Comparison with More Baselines and Variants

**Naive Baseline.** We compare SepLLM with more baselines and variants to verify its effectiveness. The results shown in Table 9 are based on a naive baseline in which $H_s$ out of $H$ attention heads are configured as sliding window attention, while the remaining heads use full attention. We use MMLU as the benchmark to measure the differences in reasoning ability, ensuring that the runtime KV usage for all methods is kept as consistent as possible. It can be observed that this naive head-wise baseline performs poorly (regardless of how many full attention heads are retained), which is due to the heterogeneity among the heads. In contrast, SepLLM achieves performance very close to Vanilla across various knowledge domains with the same KV usage.

**State-of-the-Art Baselines.** In addition, we compare SepLLM with multiple state-of-the-art baseline methods, which demonstrates the effectiveness and simplicity of SepLLM. For detailed results, please refer to the Appendix E.

**Fixed-Interval Variant.** Furthermore, to demonstrate the compression and summarization effects of separators on the information within the segments they divide, we propose

| $H_s/H$ | MMLU | | | | | $r$.KV (%) | $\boldsymbol{n}$ |
|---------|----------|--------|------|-------|---------|-----------|-----|
|         | humanity | social | stem | other | overall |           |     |
| 20/32   | 23.93 | 23.07 | 24.42 | 23.53 | 23.76 | 44.74 | 80  |
| 24/32   | 24.23 | 23.30 | 26.36 | 23.37 | 24.31 | 47.71 | 208 |
| 28/32   | 25.66 | 27.29 | 26.31 | 23.69 | 25.81 | 45.13 | 256 |
| 30/32   | 27.29 | 25.09 | 27.78 | 38.11 | 29.31 | 45.42 | 288 |
| SepLLM  | 57.66 | 76.21 | 56.49 | 72.19 | 64.68 | 44.61 | 256 |
| Vanilla | 60.49 | 76.50 | 56.61 | 72.19 | 65.72 | 100.00 | ALL |

*Table 9.* Comparison with a naive baseline method, where $H_s$ out of $H$ attention heads are set to sliding window attention, and the remaining heads use full attention. $\boldsymbol{n}$ represents the number of Neighboring Tokens (*i.e.*, the sliding window size). $r$.KV (%) indicates the ratio of KV usage at *runtime* for the respective method compared to Vanilla.

a fixed-interval variant, FixLLM. In FixLLM, instead of attending to separator tokens, it attends to one token at fixed intervals, except for the Initial Tokens' and Neighboring Tokens' sections. We find that SepLLM significantly outperforms FixLLM, which indicates that the compression and summarization capabilities of separators in SepLLM cannot be replaced by fixed-interval tokens. Please see the Appendix I for detailed results and discussions.

### 4.8. Generalization and Information Retrieval

To verify the generalization of SepLLM, we adapt SepLLM to models of different architectures and scales. The results and discussions in Appendix D can validate the generalization capability of our proposed SepLLM. Specifically, we adapt SepLLM to different backbones including Pythia-6.9B, Pythia-12B (Biderman et al., 2023), Llama-3-8B-Base/Instruct (Dubey et al., 2024) and Falcon-40B (Almazrouei et al., 2023). Moreover, we also conduct the *Needle-in-a-Haystack* experiment, which further demonstrates the compression capability of separator tokens for segment information. As illustrated in Appendix F, SepLLM can retrieve the needle in most scenarios. In comparison, StreamingLLM (Xiao et al., 2024b) cannot complete this task. In addition, we provide comparisons of SepLLM with more baseline models regarding mathematical reasoning and logical analysis capabilities in the Appendix E. Furthermore, Appendices H and I discuss in detail the effects and functions of separators from the perspective of implementation logic.

### 5. Universal Approximation

In this section, we present the theoretical results on the universal approximation capabilities of encoder-based SepLLM. Details can be found in Appendices J and K.

Let $\mathcal{T}_{\text{Sep}}^{H,d_h,d_f}$ be the class of SepLLM, where $H$, $d_h$, and $d_f$

represent the number of heads, hidden dimension in attention layers, and the hidden dimension of feed-forward layers, respectively. Denote $\mathcal{F}$ as the class of continuous functions $f : [0,1]^{d \times n} \to \mathbb{R}^{d \times n}$, where $d$ and $n$ represent the dimensionality of input tokens and the sequence length, respectively. For any $p \geq 1$, we use the $\ell_p$ distance to measure the difference between two continuous functions $f_1, f_2 \in \mathcal{F}$, defined as $\left( \int_{[0,1]^{d \times n}} \|f_1(\boldsymbol{X}) - f_2(\boldsymbol{X})\|_p^p \, d\boldsymbol{X} \right)^{1/p}$. The following theorem shows that the proposed SepLLM holds the universal approximation to arbitrarily sequence-to-sequence continuous functions. For the complete proof and derivation process, please refer to the Appendices J and K.

**Theorem 5.1.** *Given $p > 1$ and $n > 2$, for any $\epsilon > 0$ and $f \in \mathcal{F}$, there exists a SepLLM $g \in \mathcal{T}_{\text{Sep}}^{2,1,4}$, such that $d_p(f, g) < \epsilon$.*

## 6. Native Sparse Attention and Language Modeling

SepLLM essentially models sparsity based on the natural semantic distribution of natural language. During the pretraining phase, SepLLM intentionally compresses segment information into the separator used to divide the segment. This approach closely aligns with the semantic distribution of natural language because the separator itself provides a division and summary of the current segment. The segments separated out are inherently semantically coherent, forming self-contained semantic units. Therefore, **we can view SepLLM as a native sparse attention mechanism inherent to natural language itself**. Moreover, SepLLM is suitable to replace StreamingLLM (which only statically focuses on important positions and is overly sparse) as the fundamental baseline model for sparse attention mechanisms in LLMs.

## 7. Concluding Remarks

In this paper, we focus on efficient neural architecture modification to address the computational and storage challenges in LLMs, especially when processing long inputs. From the visualization of attention maps, we find that certain separator tokens consistently contribute high attention scores. Inspired by this, we propose SepLLM, a new language modeling perspective and a sparse attention mechanism, focusing attention computation on Initial, Neighboring, and Separator Tokens. To achieve wall-clock time acceleration, we also implement hardware-efficient kernels. Our training-free studies suggest these separators effectively compress segment information, enabling efficient information retrieval. Unlike previous training-free methods, SepLLM can be incorporated into the training phase, *e.g.*, training-from-scratch or post-training, thus reducing disparities between training and inference. Extensive experiments across various settings have demonstrated SepLLM's practical effectiveness.

## Impact Statement

This paper presents work whose goal is to advance the field of Machine Learning. There are many potential societal consequences of our work, none which we feel must be specifically highlighted here.

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

# Appendix

## A. Visualization of Attention Scores

We take Llama-3-8B-instruct (Dubey et al., 2024) as the model for visualization. The input sentence is "Natalia sold clips to 48 of her friends in April, and then she sold half as many clips in May. How many clips did Natalia sell altogether in April and May? Answer Natalia sold 48 clips in April. In May, she sold half as many clips as she did in April, so she sold 48/2=24 clips in May. Therefore, Natalia sold a total of 48+24=72 clips in April and May. The answer is $\boxed{72}$." and the visualization of different attention maps are shown in Figure 12,13,14.

## B. The Evolution of KV Caches

To explain the dynamic design for streaming setting better, we illustrate the detailed evolution of KV caches in Figure 7. As can be seen, $n$ and $Size_{run}$ are both periodic functions after $m_0$ tokens. And the average KV cache usage is much less than the maximum capacity $c$.

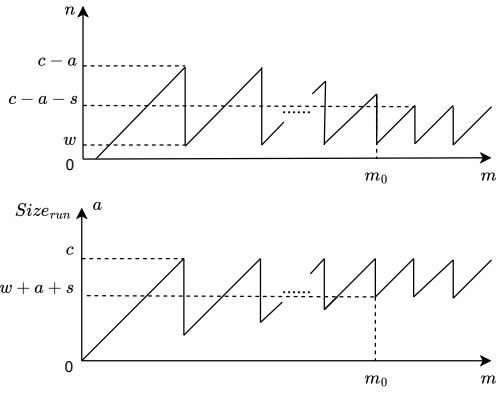

*Figure 7.* The evolution of KV caches in the streaming setting.

## C. Training Acceleration

We list the detailed wall-clock time per iteration and throughput in Table 10. The speed-up ratio is around 1.53.

| | Vanilla (Full Attention) | SepLLM (n=64) | SepLLM (n=128) |
|---|---|---|---|
| time per iteration (ms) | 2524.45 | 1648.11 | 1653.11 |
| samples / second | 405.82 | 622.31 | 620.30 |

*Table 10.* The details about training acceleration.

We can see that the speeds of *SepLLM(n*=128) and *SepLLM(n*=64) are almost the same, which is attributed to the excellent parallelization capability of *Sep-Attention* module.

## D. The Performance of Different Models

**Different Architectures.**  Concerning different decoder-only models, we test our SepLLM on Llama3 and Pythia backbones on PG19 test dataset (generating 64K tokens). The results are shown in Table 11 (*a*=4, *c*=800 for both SepLLM and StrmLLM. *s*=64,*w*=256 for SepLLM).

| Backbone | Arch. | $c$ | $r$.KV | ppl | time(s) |
|---|---|---|---|---|---|
| | Vanilla | 64K | 32K | 1037.6 | 4160.7 |
| Pythia-6.9B | StrmLLM | 800 | 800 | 15.9 | 1522.6 |
| | SepLLM | 800 | 562 | 15.8 | 1456.0 |
| | Vanilla | 64K | 32K | 1090.8 | 3380.6 |
| Llama-3-8B | StrmLLM | 800 | 800 | 37.9 | 1096.0 |
| | SepLLM | 800 | 562 | 33.4 | 1049.7 |

*Table 11.* The comparison of SepLLM adapted to different architectures.

From the above table, it can be seen that for models with similar size, setting a similar KV retention rate can yield similarly good performance.

**Different Scales.**  To learn the generalization to different scales, we test our SepLLM on Pythia-6.9B and Pythia-12B backbones on PG19 test dataset (generating 20K tokens). The results are illustrated in Table 12.

| Backbone | $a$ | $s$ | $w$ | $c$ | $r$.KV | ppl | time(s) |
|---|---|---|---|---|---|---|---|
| | 4 | 64 | 256 | 800 | 562 | 13.0 | 445.0 |
| Pythia-6.9B | 4 | 64 | 800 | 1024 | 946 | 12.7 | 450.4 |
| | 4 | 64 | 928 | 1280 | 1138 | 12.7 | 454.4 |
| Pythia-12B | 4 | 64 | 256 | 800 | 562 | 12.1 | 577.0 |

*Table 12.* The comparison of SepLLM adapted to Pythia (Biderman et al., 2023) with different scales.

Compared to Pythia-12B, the smaller model Pythia-6.9B will have a higher perplexity if the entire KV cache capacity is the same (*c*=800). Therefore, it is necessary to increase *c* to achieve a lower perplexity close to that of Pythia-12B (*c*=800). On the other hand, the larger models will have lower perplexity but require a longer inference time.

**Larger Models**  Falcon-40B (Almazrouei et al., 2023) is another larger architecture we adapted to evaluate

the scalability of our proposed SepLLM. The experiment results are shown in Table 13, where we set *a*=4,*s*=64,*w*=512/720, *c*=800/1024 for SepLLM, and *a*=4, *c*=800/1024 for StreamingLLM. And the conclusions are similar to the previous parts.

| Length | Methods | *c* | *r*.KV | ppl | time (s) |
|--------|---------|-----|--------|-----|----------|
| 20K | StrmLLM | 1024 | 1024 | 8.98 | 1512.88 |
| | StrmLLM | 800 | 800 | 9.02 | 1430.59 |
| | SepLLM | 1024 | 906 | 8.92 | 1440.89 |
| | SepLLM | 800 | 690 | 9.00 | 1368.07 |
| 64K | StrmLLM | 1024 | 1024 | 11.01 | 4844.79 |
| | StrmLLM | 800 | 800 | 11.09 | 4623.90 |
| | SepLLM | 1024 | 906 | 10.96 | 4619.63 |
| | SepLLM | 800 | 690 | 11.07 | 4414.72 |

*Table 13.* The comparison of SepLLM adapted to Falcon-40B (Almazrouei et al., 2023).

**Base or Instruct.** In general, whether it is the base model or the instruction-tuned model, we can condense the segment information into the corresponding Key-Value pairs of the separator tokens. To illustrate, we fine-tune Llama-3-8B-instruct and Llama-3-8B-base models (Dubey et al., 2024) on LongAlpaca dataset (Chen et al., 2024) for only 200 and 500 steps, respectively. After fine-tuning, we take GSM8K-CoT (Cobbe et al., 2021) as the benchmark for reasoning ability evaluation. We find that both base and instruction-tuned models exhibit excellent performance (matching or surpassing the vanilla model with original attention mechanism). The only difference is that for Llama-3-8B-base, we need to fine-tune for more steps to achieve such performance. In comparison, Llama-3-8B-instruct requires fewer fine-tuning steps. Even in a training-free scenario, Llama-3-8B-instruct demonstrates decent performance. This indicates that the embeddings of Llama-3-8B-instruct align with the distribution required by the SepLLM architecture better.

| Backbone | Algorithm | GSM8K-CoT | *r*.KV (%) |
|----------|-----------|-----------|------------|
| Base | Vanilla | 54.44 | 100 |
| | SepLLM ft. | 55.95 | 47.36 |
| Instruct | Vanilla | 77.26 | 100 |
| | SepLLM ft. | 77.63 | 47.36 |

*Table 14.* The comparison of SepLLM adapted to Llama-3-8B (Dubey et al., 2024) of base or instruct versions.

## E. Extended Comparisons

We use GSM8K-CoT (Cobbe et al., 2021), the most commonly used metric for testing mathematical reasoning and logical analysis, to compare SepLLM (*a*=3, *n*=256) with other state-of-the-art training-free methods, including H$_2$O (Zhang et al., 2023), SnapKV (Li et al., 2024), and

| | flexible-extract | strict-match | *r*.KV (%) |
|--------|------------------|--------------|------------|
| Vanilla | 77.79 | 77.26 | 100.00 |
| SepLLM | 77.18 | 77.18 | 47.36 |
| H2O | 76.27 | 75.06 | 47.54 |
| SnapKV | 76.50 | 73.62 | 47.54 |
| PyramidKV | 75.82 | 72.02 | 47.54 |

*Table 15.* Evaluation results and average *runtime* KV cache usage for experiments on GSM8K-CoT with 8-shots, compared to multiple baseline methods.

PyramidKV (Zhang et al., 2024). All methods are configured to retain nearly identical runtime KV cache usage. The results are shown in Table 15. It can be observed that SepLLM, without requiring complex importance evaluation mechanisms, is able to maintain strong reasoning capabilities simply by compressing segment information.

## F. Needle in a Haystack

To evaluate the long-context information retrieval ability of our proposed SepLLM, we take *Needle in a Haystack*[2] as the benchmark and compare the performance of SepLLM and StreamingLLM. The results are shown in Figure 8,9,10,11 and SepLLM can achieve more scores compared to StreamingLLM. This experiment indeed validates that SepLLM can effectively compress information of segments into the KV corresponding to separators, as even though the KV corresponding to tokens in the needle (except for possibly existing separators) are discarded, SepLLM can still retrieve the needle.

## G. Choice of Separators

In this paper, we treat the choice of separators as a hyperparameter. Since the number of commonly used separators is typically limited in most languages, we recommend including all types of separators as much as possible. In Table 16, we present the experimental results based on different choices of separators. We use the GSM8K-CoT metric to reflect the performance differences caused by these choices (GSM8K-CoT is a commonly used metric for evaluating the mathematical reasoning and logical analysis capabilities of LLMs). It can be observed that when using four common separators instead of the default nine, the performance shows a certain degree of decline. Furthermore, when only "." and "?" are used, the model's reasoning ability drops significantly. When all separators are removed, this performance degradation becomes even more pronounced (*i.e.*, *StrmLLM(n=256)*). While increasing the number of Neighboring Tokens (*n*) to ensure the total amount of retained KV cache remains unchanged can partially mitigate the per-

---

[2]https://github.com/gkamradt/LLMTest_NeedleInAHaystack

formance loss (*i.e.*, *StrmLLM(n=380)*), it falls far short of restoring the performance to the levels achieved with the default nine separators or the four common separators.

## H. Discussions on Separators

We provide the following assumptions and discussions on why keeping the KV corresponding to separators can maintain the performance of the original model.

**Training from Scratch**    During our training process, we enforce that each current token can only see its preceding neighboring tokens, separator tokens, and initial tokens such that the model is compelled to condense the information of each segment into the Key-Value pairs corresponding to the separators through the self-attention mechanism. Therefore, the hidden embedding of a separator is functionally similar to the state space of an RNN, even though the computation method differs as we utilize the attention mechanism. Furthermore, since the length of each segment is typically finite, short and balanced (He et al., 2024), the compressed information is efficient and less likely to be forgotten.

**Training-Free**    First, these separators (commas, periods) are extremely high-frequency tokens (both in the pre-training text and in the text generated by the language model). Thus, during the pre-training process, these separators are the most common context for all other tokens in the vocabulary. Consequently, their embeddings exhibit greater similarity, resulting in larger attention values when multiplied with other tokens. Furthermore, from a logical standpoint, it is evident that separators need to be generated by the language model very frequently. Therefore, their attention values with respect to any other token cannot be too small; otherwise, they would not be generated frequently by the language model. Tokens with larger mutual attention values will incorporate more information from the other tokens. Hence, from a semantic perspective, generating a separator serves as a summarization of the current segment, naturally dividing and summarizing the contextual semantics.

## I. Fixed-Interval Variant

Sparsity is ubiquitous across various types of neural networks (Shi et al., 2021; Chen et al., 2025a; Zheng et al., 2025; Yuan et al., 2025; Chen et al., 2025b). To verify the sparsity caused by separators as natural semantic divisions in natural language and the resulting importance of these separators, we propose another variant, *i.e.*, when calculating sparse attention, instead of focusing only on Separator Tokens between the Initial Tokens and Neighboring Tokens, we attend to one token at fixed intervals (e.g., every 8 tokens or 16 tokens), while the other tokens are masked, namely *FixLLM*. We evaluated the mathematical reasoning ability and knowledge-based reasoning ability of

the two variants using the GSM8K-CoT (Cobbe et al., 2021) and MMLU (Hendrycks et al., 2021)) benchmarks under a training-free setting, based on the Llama3-8B-Instruct backbone. The results in Table 17 show that FixLLM has a significant gap compared to SepLLM in both mathematical logical reasoning and knowledge-based reasoning capabilities. Therefore, the summarization and compression effects of separators in SepLLM for the segments they divide cannot be replaced by fixed-interval tokens.

## J. Universal Approximation

In this section, we theoretically analyze the universal approximation capabilities of encoder-based SepLLM. Let $\mathcal{T}_{\text{Sep}}^{H,d_h,d_f}$ be the class of SepLLM, where $H$, $d_h$, and $d_f$ represent the number of heads, hidden dimension in attention layers, and the hidden dimension of feed-forward layers, respectively. In the attention layer of SepLLM, token $i$ can attend to its neighboring tokens with a sliding window of size $\ell$ (i.e., tokens in the range $[i - \ell, i + \ell]$) and all special tokens of the sequence. Additionally, we assume that for at most $s$ successive tokens, a special token will appear in the sequence. Denote $\mathcal{F}$ as the class of continuous functions $f : [0,1]^{d \times n} \to \mathbb{R}^{d \times n}$, where $d$ and $n$ represent the dimensionality of input tokens and the sequence length, respectively. For any $p \geq 1$, we use the $\ell_p$ distance to measure the difference between two continuous functions $f_1, f_2 \in \mathcal{F}$, defined as $\left( \int_{[0,1]^{d \times n}} \|f_1(\boldsymbol{X}) - f_2(\boldsymbol{X})\|_p^p \, d\boldsymbol{X} \right)^{1/p}$. The following theorem shows that the proposed SepLLM holds the universal approximation to arbitrarily sequence-to-sequence continuous functions.

**Theorem J.1.** *Given $p > 1$ and $n > 2$, for any $\epsilon > 0$ and $f \in \mathcal{F}$, there exists a SepLLM $g \in \mathcal{T}_{Sep}^{2,1,4}$, such that $d_p(f, g) < \epsilon$.*

We outline the key steps of the proof here, with the full details provided later. The proof follows the approach in Yun et al. (2020).

**Step 1**: We begin by dividing the region $[0,1]^{d \times n}$ into a set of grid points $\mathcal{G}_\delta = \{0, \delta, 2\delta, \ldots, 1\}^{d \times n}$, where each point in $[0,1]^{d \times n}$ corresponds to a cube defined by these grid points. Here, $\delta > 0$ determines the resolution of grid points. Specifically, for any $\boldsymbol{X} \in [0,1]^{d \times n}$, there exists a grid point $\boldsymbol{X}_\delta \in \mathcal{G}_\delta$, such that $\boldsymbol{X}$ lies within the cube $\boldsymbol{X}_\delta + [0, \delta]^{d \times n}$. We then assign the same function values to all inputs belonging to the same cube, as determined by a piecewise constant function $\bar{f}$. For any $\epsilon > 0$, there exists a sufficiently small $\delta > 0$, such that $d_p(f, \bar{f}) \leq \frac{\epsilon}{2}$.

**Step 2**: We replace the softmax and ReLU activation in attention layers and feed-forward layers of SepLLM by the hardmax operator (i.e., $\arg\max$) and piecewise linear functions (at most three pieces). We denote the class of

| | SepLLM (n=256) | SepLLM ($n$=256) | SepLLM ($n$=256) | StrmLLM ($n$=256) | StrmLLM ($n$=380) |
|---|---|---|---|---|---|
| Separators | ".", ";", "?", "!", ",", ":", " ", "\t", "\n" | ",", ".", "?", ";" | ".", "?" | None | None |
| $r$.KV (%) | 47.36 | 37.92 | 36.44 | 31.92 | 47.54 |
| flexible-extract | 77.18 | 76.68 | 70.66 | 68.84 | 71.42 |
| strict-match | 77.18 | 76.85 | 70.36 | 67.63 | 70.89 |

*Table 16.* Evaluation results and average *runtime* KV cache usage for training-free experiments on GSM8K-CoT with 8-shots, based on different choices of separators.

| | GSM8K-CoT | | | MMLU | | | | | |
|---|---|---|---|---|---|---|---|---|---|
| | flexible | strict | $r$.KV (%) | humanities | stem | social | other | overall | $r$.KV (%) |
| Vanilla | 77.79 | 77.26 | 100.00 | 60.49 | 56.61 | 76.50 | 72.19 | 65.72 | 100.00 |
| FixLLM ($\Delta$=5, $n$=256) | 70.43 | 70.43 | 45.64 | 55.52 | 54.80 | 72.99 | 69.75 | 62.33 | 50.20 |
| FixLLM ($\Delta$=4, $n$=256) | 72.71 | 72.33 | 49.08 | 55.92 | 54.39 | 74.36 | 70.81 | 62.91 | 53.32 |
| SepLLM ($n$=256) | 77.18 | 77.18 | 47.36 | 57.66 | 56.49 | 76.21 | 72.19 | 64.68 | 44.61 |

*Table 17.* Evaluation results and average *runtime* KV cache usage for training-free experiments on GSM8K-CoT 8-shots and MMLU 5-shots. For SepLLM and FixLLM, three initial tokens' KV are kept. $\Delta$ denotes the interval size for FixLLM and $n$ is the number of retained neighboring tokens' KV. $r$.KV (%) represents the ratio of KV usage at *runtime* for the respective method compared to Vanilla.

the modified SepLLM models by $\bar{\mathcal{T}}_{\text{Sep}}^{H,d_h,d_f}$. For the above piecewise linear function $\bar{f}$, there exists a modified SepLLM $\bar{g} \in \bar{\mathcal{T}}_{\text{Sep}}^{2,1,1}$, such that $\bar{g} = \bar{f}$.

**Step 3**: Finally, we approximate the modified SepLLM $\bar{g} \in \bar{\mathcal{T}}_{\text{Sep}}^{2,1,1}$ by a standard SepLLM $g \in \mathcal{T}_{\text{Sep}}^{2,1,4}$, i.e., we have $d_p(\bar{g}, g) < \frac{\epsilon}{2}$. This approximation is justified by the fact that the softmax function can approximate the hardmax operator arbitrarily closely when the temperature parameter is sufficiently large. Additionally, feed-forward networks with ReLU activation can effectively represent any piecewise linear function.

## K. Proof for Theorem J.1

**Lemma K.1** (Lemma 5 in Yun et al. (2020)). *For any $f \in \mathcal{F}$, $\epsilon > 0$, and $p \geq 1$, there exists a piecewise constant function $\bar{f}$, such that $d_p(f, \bar{f}) < \frac{\epsilon}{2}$.*

To identify the position of tokens in SepLLM, we include the position encoding $\boldsymbol{E} \in \mathbb{R}^{d \times n}$ into the token $\boldsymbol{X}$, i.e., the input token is $\boldsymbol{X} + \boldsymbol{E}$. Here, for theoretical convenience, the positional encoding matrix is defined as

$$\boldsymbol{E} = [(n-1)\mathbf{1}, \mathbf{0}, \mathbf{1}, \dots, (n-2)\mathbf{1}].$$

With this encoding, the input token lies within the range $\boldsymbol{X} + \boldsymbol{E}$ takes in the range $[0, n]^{d \times n}$. The input token is then mapped to grid points using a quantization function $g_q$, which can be implemented using feed-forward neural networks with piecewise linear activation functions (i.e., the feed-forward layers of modified SepLLM).

**Lemma K.2** (Lemma 6 in Yun et al. (2020)). *Consider the*

*quantization mapping $g_q^{ent}$:*

$$g_{\text{q}}^{\text{ent}}(t) = \begin{cases} k\delta, & \text{if } k\delta \leq t < (k+1)\delta, \quad k \in [0 : n/\delta - 1], \\ t, & \text{otherwise.} \end{cases}$$

(5)

*There exists a modified SepLLM with feed-forward layers and identity attention layers, $g_q \in \bar{\mathcal{T}}_{Sep}^{2,1,1}$ composed of $\frac{nd}{\delta}$ layers with $n_f = 1$ and piecewise linear activation functions, such that $g_q$ realizes the quantization mapping $g_q^{ent}$.*

Let $\boldsymbol{Z} = g_q(\boldsymbol{X} + \boldsymbol{E})$ denote the quantized input token matrix, which corresponds to mapping $\boldsymbol{X} + \boldsymbol{E}$ to the grid point at the leftmost corner of the cube. Denote $\mathcal{G}_\delta$ as the set of all quantized token matrices derived from $\boldsymbol{X} + \boldsymbol{E}$. The following definition of contextual mapping aims at uniquely distinguishing each grid point (i.e., quantized tokens) from all possible combinations.

**Definition K.3** (Contextual Mapping). For a given set of grid points $\mathcal{G}_\delta \subset \mathbb{R}^{d \times n}$, a contextual mapping $q : \mathcal{G}_\delta \to \mathbb{R}^n$ satisfies:

- For any $\boldsymbol{G} \in \mathcal{G}_\delta$, all entries in $q(\boldsymbol{G})$ are distinct.

- For any $\boldsymbol{G}, \boldsymbol{G}' \in \mathcal{G}_\delta$ with $\boldsymbol{G} \neq \boldsymbol{G}'$, all entries of $[q(\boldsymbol{G}), q(\boldsymbol{G}')] \in \mathbb{R}^{2n}$ are distinct.

**Lemma K.4.** *There exists a function $g_c \in \bar{\mathcal{T}}_{Sep}^{2,1,1} : \mathbb{R}^{d \times n} \to \mathbb{R}^{d \times n}$, consisting of $\frac{n-1}{\delta^d} + \lceil \frac{s}{\ell} \rceil$ layers of modified SepLLM (purely attention layers with identity feed-forward layers), such that $q(\boldsymbol{Z}) := \boldsymbol{u}^\top g_c(\boldsymbol{Z})$ is a contextual mapping. Here, the modified SepLLM means replacing all softmax functions with hardmax operators.*

*Proof.* Denote $i$-th column of $\boldsymbol{Z}$ as $\boldsymbol{Z}_i$ and let $\boldsymbol{u} = \left(1, \delta^{-1}, \delta^{-2}, \dots, \delta^{-d+1}\right)^\top$. Based on the definition of the

quantization mapping $g_q$ and the selection of positional encoding matrix $\boldsymbol{E}$, we have

$$\boldsymbol{u}^\top \boldsymbol{Z}_1 \in$$

$$\left[(n-1)\sum_{i=0}^{d-1}\delta^{-i} : \delta : (n-1)\sum_{i=0}^{d-1}\delta^{-i} + \delta^{-d+1} - \delta\right],$$

$$\boldsymbol{u}^\top \boldsymbol{Z}_k \in$$

$$\left[(k-2)\sum_{i=0}^{d-1}\delta^{-i} : \delta : (k-2)\sum_{i=0}^{d-1}\delta^{-i} + \delta^{-d+1} - \delta\right],$$

(6)

for $k \in [2:n]$. We observe that those intervals corresponding to distinct $k$ are disjoint. Denote $\boldsymbol{u}^\top \boldsymbol{Z}_k$ as $z_k$, it follows that $z_2 < z_3 < \ldots < z_n < z_1$. We define the operator

$$\Psi(\boldsymbol{Z}; b)_k$$
$$= \boldsymbol{u}^T \boldsymbol{Z}_{\mathcal{A}_k} \sigma_{\mathrm{H}}\left[\left(\boldsymbol{u}^T \boldsymbol{Z}_{\mathcal{A}_k}\right)^T \left(\boldsymbol{u}^T \boldsymbol{Z}_k - b\right)\right]$$
$$= \begin{cases} \max_{j \in \mathcal{A}_k} \boldsymbol{u}^T \boldsymbol{Z}_j & \text{if } \boldsymbol{u}^T \boldsymbol{Z}_k > b, \\ \min_{j \in \mathcal{A}_k} \boldsymbol{u}^T \boldsymbol{Z}_j & \text{if } \boldsymbol{u}^T \boldsymbol{Z}_k < b, \end{cases}$$

where $\mathcal{A}_k$ denotes the set of tokens that the token $k$ can attend to in SepLLM, consisting of its neighboring tokens and special tokens. The operator $\Psi(\boldsymbol{Z}; b)_k$ can be implemented using a one-head attention layer in the modified SepLLM with the hardmax operator as the activation function. We further define the following operator, which can be implemented using a two-head attention layer in the modified SepLLM:

$$\Phi(\boldsymbol{Z}; c; b_{\min}, b_{\max})_k$$
$$= \boldsymbol{Z} + c\left(\Psi(\boldsymbol{Z}; b_{\max})_k - \Psi(\boldsymbol{Z}; b_{\min})_k\right)\boldsymbol{e}_1$$
$$= \begin{cases} \boldsymbol{Z} + c\left(\max_{j \in \mathcal{A}_k} \boldsymbol{u}^T \boldsymbol{Z}_j - \min_{j \in \mathcal{A}_k} \boldsymbol{u}^T \boldsymbol{Z}_j\right)\boldsymbol{e}_1 \\ \boldsymbol{Z} \end{cases}.$$

Here, the first condition is satisfied if $b_{\min} < \boldsymbol{u}^T \boldsymbol{Z}_k < b_{\max}$, and the second condition applies otherwise. For $k = 2$ (the second token), we apply the operator $\Phi(\boldsymbol{Z}; \delta^{-d}; b - \frac{\delta}{2}, b + \frac{\delta}{2})$ for $\delta^{-d}$ times, with $b$ varying in the range $[0 : \delta : \delta^{-d+1} - \delta]$. Note that the operator modifies only the $k$-th token while leaving all other tokens unchanged, because $\boldsymbol{u}^\top \boldsymbol{Z}_k$ lies in disjoint intervals. After this operation, we denote the updated second token as $\widetilde{\boldsymbol{Z}}_2$, and we have

$$\widetilde{\boldsymbol{Z}}_2 = \boldsymbol{Z}_2 + \delta^{-d}(z_1 - z_2)\boldsymbol{e}_1,$$

and

$$\tilde{z}_2 := \boldsymbol{u}^\top \widetilde{\boldsymbol{Z}}_2 = z_2 + (z_1 - z_2)\delta^{-d} > z_1.$$

Similarly, Similarly, for the third token ($k = 3$), we apply the operator $\Phi(\boldsymbol{Z}; \delta^{-d}; b - \frac{\delta}{2}, b + \frac{\delta}{2})$ with $b$ varying in the range $\left[\sum_{i=0}^{d-1}\delta^{-i} : \delta : \sum_{i=0}^{d-1}\delta^{-i} + \delta^{-d+1} - \delta\right]$. This operation modifies only the third token (the third column of

matrix $\boldsymbol{Z}$), resulting in:

$$\widetilde{\boldsymbol{Z}}_3 = \boldsymbol{Z}_3 + \delta^{-d}(\tilde{z}_2 - z_3)\boldsymbol{e}_1,$$

and

$$\tilde{z}_3 := \boldsymbol{u}^\top \widetilde{\boldsymbol{Z}}_3 = z_3 + (\tilde{z}_2 - z_3)\delta^{-d} > \tilde{z}_2.$$

We repeat this process for the remaining tokens until the last token $k = n$. As a result, the obtained token matrix $\widetilde{\boldsymbol{Z}}$ satisfies:

$$\tilde{z}_1 < \tilde{z}_2 < \ldots < \tilde{z}_n,$$

where $\tilde{z}_k := \boldsymbol{u}^\top \widetilde{\boldsymbol{Z}}_k$. In summary, there exists a modified SepLLM with $\frac{n-1}{\delta^d}$ layers capable of transforming the token matrix $\boldsymbol{Z}$ into $\widetilde{\boldsymbol{Z}}$.

Now, we prove that the mapping from $\boldsymbol{Z}$ to $\tilde{z}_n$ is injective. Suppose there exist two token matrices $\boldsymbol{Z}, \boldsymbol{Z}' \in \mathcal{G}_\delta$, such that $\tilde{z}_n = \tilde{z}'_n$. By induction, we have

$$\tilde{z}_n = z_n + \delta^{-d}\left(\tilde{z}_n - z_n\right)$$
$$= \cdots$$
$$= z_n + \sum_{i=1}^{n-1}\delta^{-id}(z_{n-i} - z_{n+1-i}).$$

If $z_n \neq z'_n$, $|z_n - z'_n| \leq \delta^{-d+1} - \delta$. However, the term $\sum_{i=1}^{n-1}\delta^{-id}(z_{n-i} - z_{n+1-i})$ is dominant, and cannot cancel $|z_n - z'_n|$. If $z_n = z'_n$ but $z_{n-1} \neq z'_{n-1}$, we encounter a similar contradiction. Since each term in the sum has a different scale of $\delta^{-1}$, $\tilde{z}_n = \tilde{z}'_n$ holds if and only if $\boldsymbol{Z} = \boldsymbol{Z}'$.

For the current token matrix $\widetilde{\boldsymbol{Z}} = \left[\widetilde{\boldsymbol{Z}}_1, \widetilde{\boldsymbol{Z}}_2, \ldots, \widetilde{\boldsymbol{Z}}_n\right]$, the dominant term for each column is located in the first row, where

$$\left(\widetilde{\boldsymbol{Z}}_k\right)_1 = (\boldsymbol{Z}_k)_1 + \delta^{-d}(\tilde{z}_{k-1} - z_k).$$

Moreover, the dominant term for the first element of each column is $\delta^{-d}\tilde{z}_{k-1}$. Since $\tilde{z}_n$ can be viewed as the identity of the original token matrix $\boldsymbol{Z}$, the last column, which depends on $\widetilde{z}_n$, is distinct for different token matrices. However, for the other columns dominated by $\widetilde{z}_k$ ($k \neq n$), the uniqueness is not guaranteed. To address this, we apply a series of token transmission steps, propagating information from the last token to all other tokens. The core idea relies on the fact that the last token can attend to the nearest special tokens with the help of neighboring tokens, requiring at most $\lceil \frac{s}{\ell} \rceil$ layers. Following Section E.2.4 in (Yun et al., 2020), after $\lceil \frac{s}{\ell} \rceil$ layers, $\widetilde{z}_n$ is successfully copied to all tokens. As a result, each column of the updated token matrix, denoted as $\widetilde{\widetilde{\boldsymbol{Z}}}$, depends on $\widetilde{z}_n$. Furthermore, all elements of $\boldsymbol{u}^\top \widetilde{\widetilde{\boldsymbol{Z}}}$ are distinct and lie in disjoint intervals. Since $\tilde{z}_n$ acts as an identity to the token matrix $\boldsymbol{Z}$, the mapping $\boldsymbol{u}^\top \widetilde{\widetilde{\boldsymbol{Z}}}$ satisfies all conditions of contextual mappings. The mapping from $\boldsymbol{Z}$ to $\widetilde{\widetilde{\boldsymbol{Z}}}$ can be realized by $\frac{n-1}{\delta^d} + \lceil \frac{s}{\ell} \rceil$ layers of a

modified SepLLM, consisting of purely attention layers and identity feed-forward layers. □

**Lemma K.5** (Lemma 8 in Yun et al. (2020)). *For the contextual mapping $g_c$ in Lemma K.4, there exist $\frac{n}{\delta^{dn}}$-layer modified SepLLM (composed of feed-forward layers and identity attention layers), denoted as $g_v \in \bar{\mathcal{T}}_{Sep}^{2,1,1} : \mathbb{R}^{d \times n} \to \mathbb{R}^{d \times n}$, such that*

$$g_v(g_c(\boldsymbol{Z}))_k = \bar{f}(\boldsymbol{Z} - \boldsymbol{E})_k,$$

*where $k$ denotes the $k$-th column of the token matrix.*

From the above analysis, we obtain

$$\bar{g}(\boldsymbol{X}) = g_v \circ g_c \circ g_q(\boldsymbol{X} + \boldsymbol{E}) = \bar{f}(\boldsymbol{X}),$$

where the quantization mapping $g_q$, the contextual mapping $g_c$ and the value mapping $g_v$ are all realized by the modified SepLLM. Here $\bar{g}$ represents the modified SepLLM with $\frac{nd}{\delta} + \frac{n-1}{\delta^d} + \lceil \frac{s}{\ell} \rceil + \frac{n}{\delta^{nd}}$ layers. The following lemma states that the modified SepLLM can be approximated by a standard SepLLM with arbitrary small error.

**Lemma K.6** (Lemma 4 in Yun et al. (2020)). *For any modified SepLLM $\bar{g} \in \bar{\mathcal{T}}_{Sep}^{2,1,1}$ and any $\epsilon > 0$, there exists a SepLLM $g \in \mathcal{T}_{Sep}^{2,1,4}$, such that $d_p(g, \bar{g}) < \epsilon$.*

By combining all the lemmas above, we conclude that for any $f \in \mathcal{F}$ and $\epsilon > 0$, there exists a SepLLM $g \in \mathcal{T}_{Sep}^{2,1,4}$, such that $d_p(f, g) < \epsilon$.

## L. AI Assistant

We only use AI assistants (*e.g.*, GPT-3.5-Turbo) to polish and refine the article.

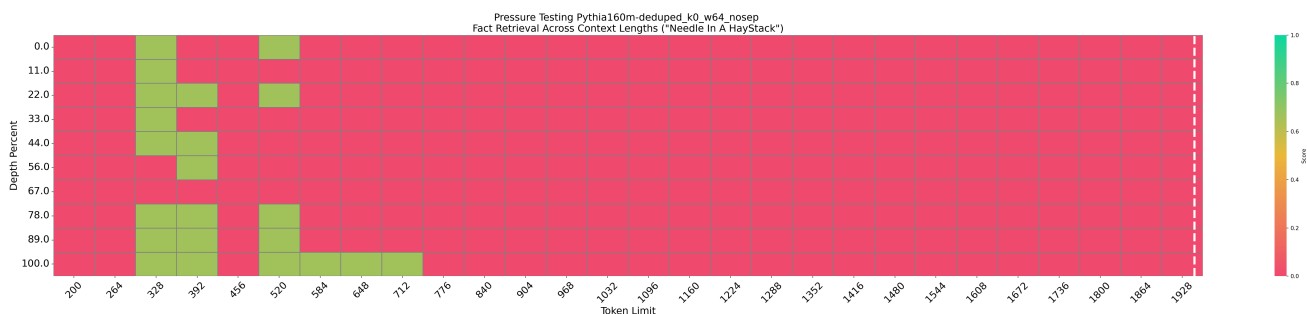

*Figure 8. Needle-in-a-Haystack* test results for StreamingLLM (***n*=64**) based on Pythia-160M-deduped.

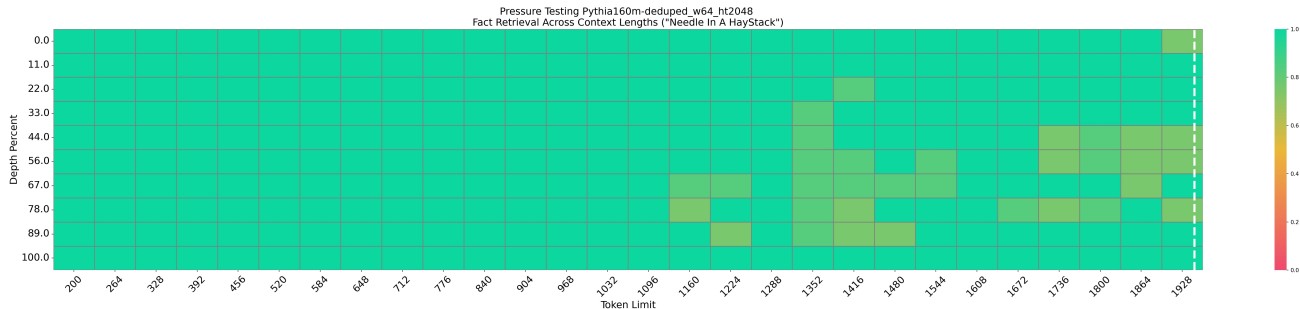

*Figure 9. Needle-in-a-Haystack* test results for our SepLLM(***n*=64**, H/T) based on Pythia-160M-deduped.

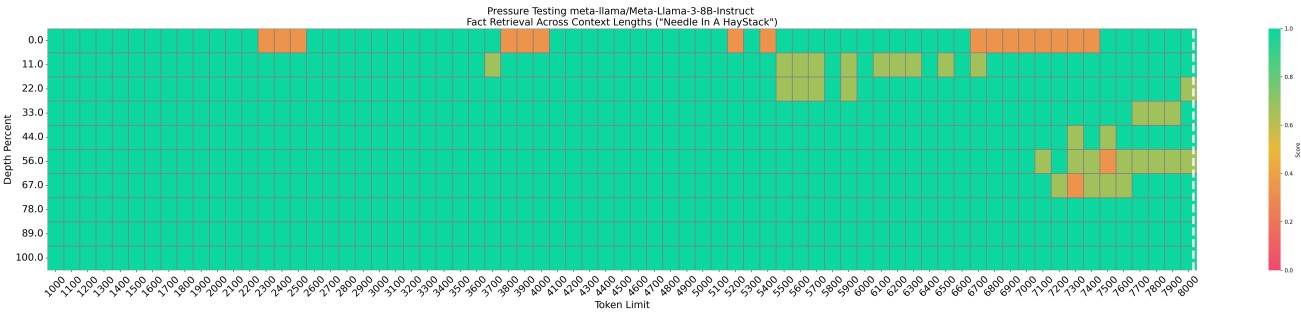

*Figure 10. Needle-in-a-Haystack* test results for our SepLLM(***n*=2048**; first/last 2 layers (4 layers in total): full attention) based on Llama-3-8B-instruct. 4 initial tokens are kept.

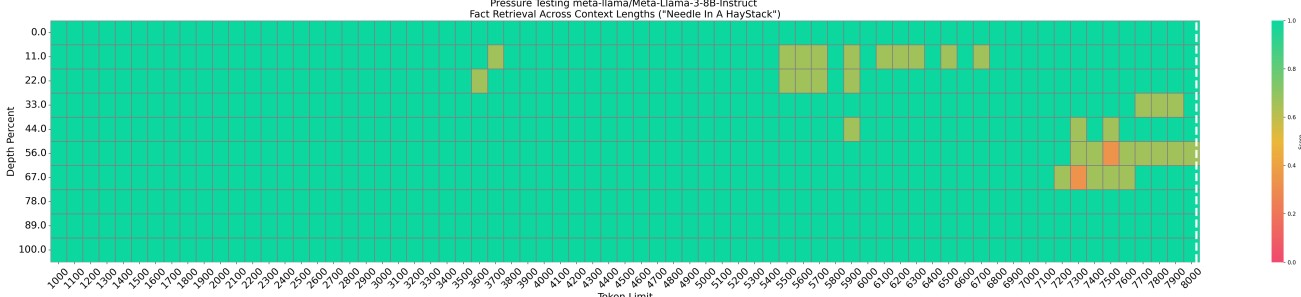

*Figure 11. Needle-in-a-Haystack* test results for our SepLLM(***n*=2048**; first/last 2 layers (4 layers in total): full attention) based on Llama-3-8B-instruct. 32 initial tokens are kept.

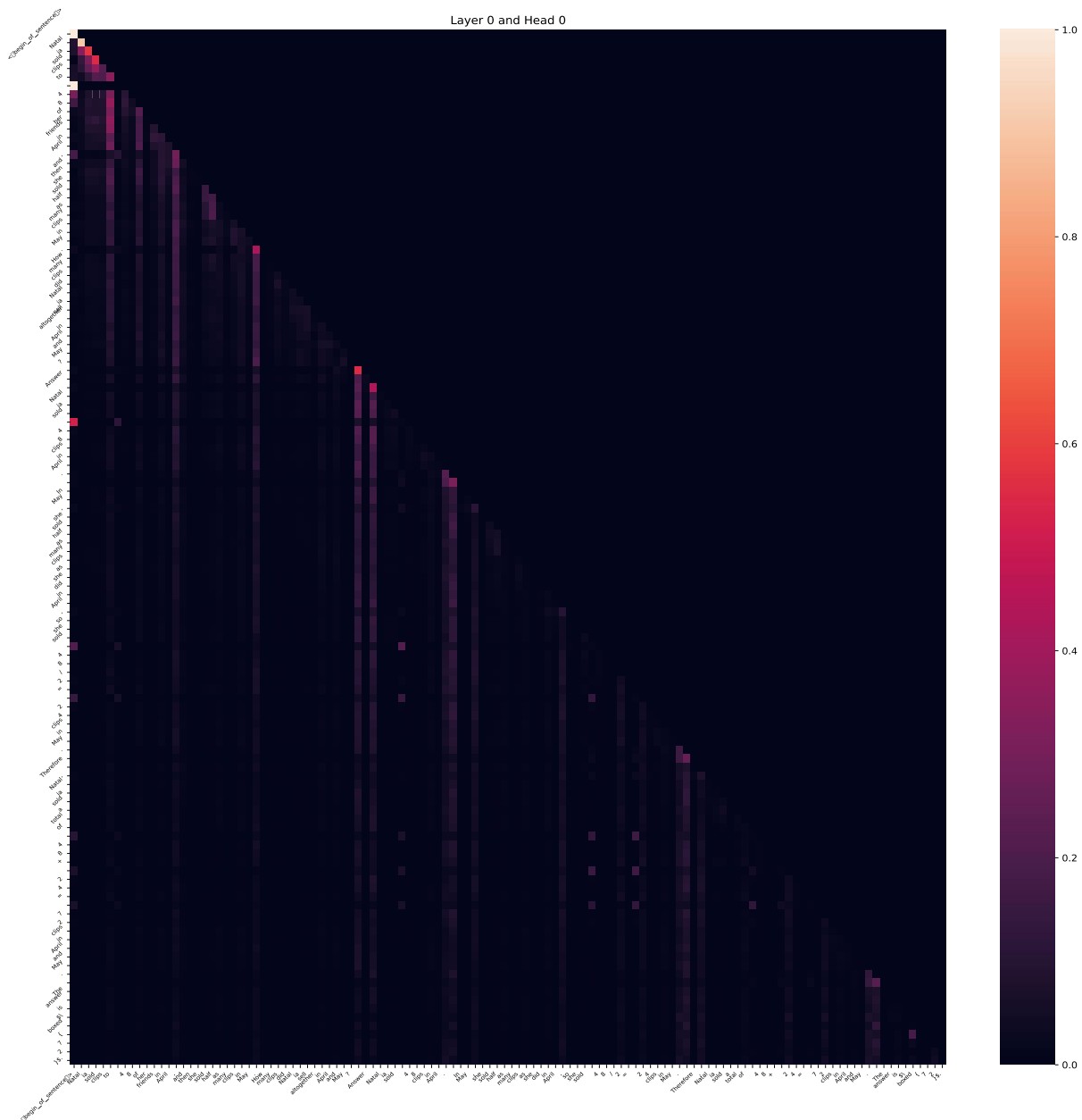

*Figure 12.* An example of attention map in Llama-3-8B-Instruct (Layer 0 and Head 0).

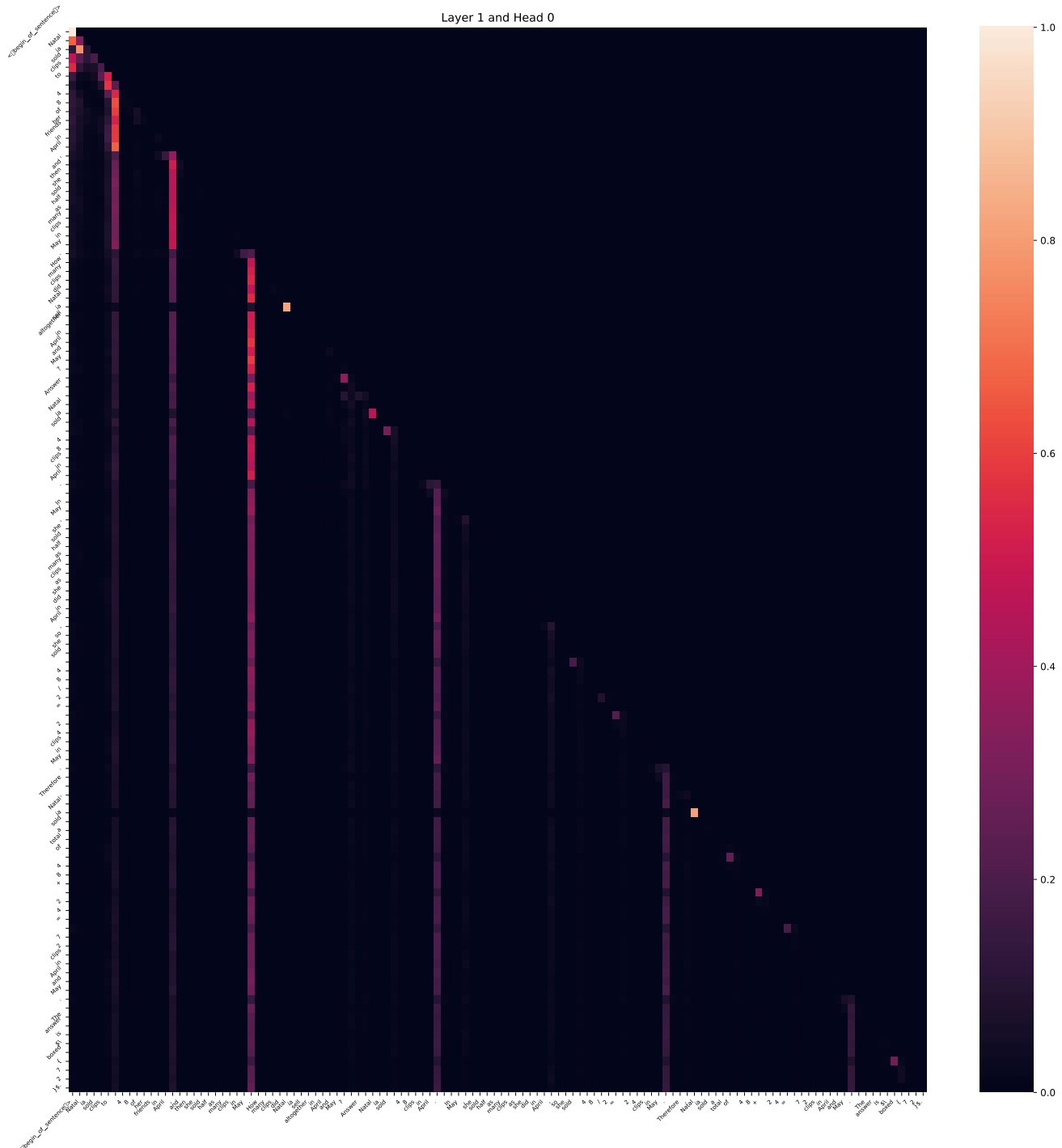

*Figure 13.* An example of attention map in Llama-3-8B-Instruct (Layer 1 and Head 0).

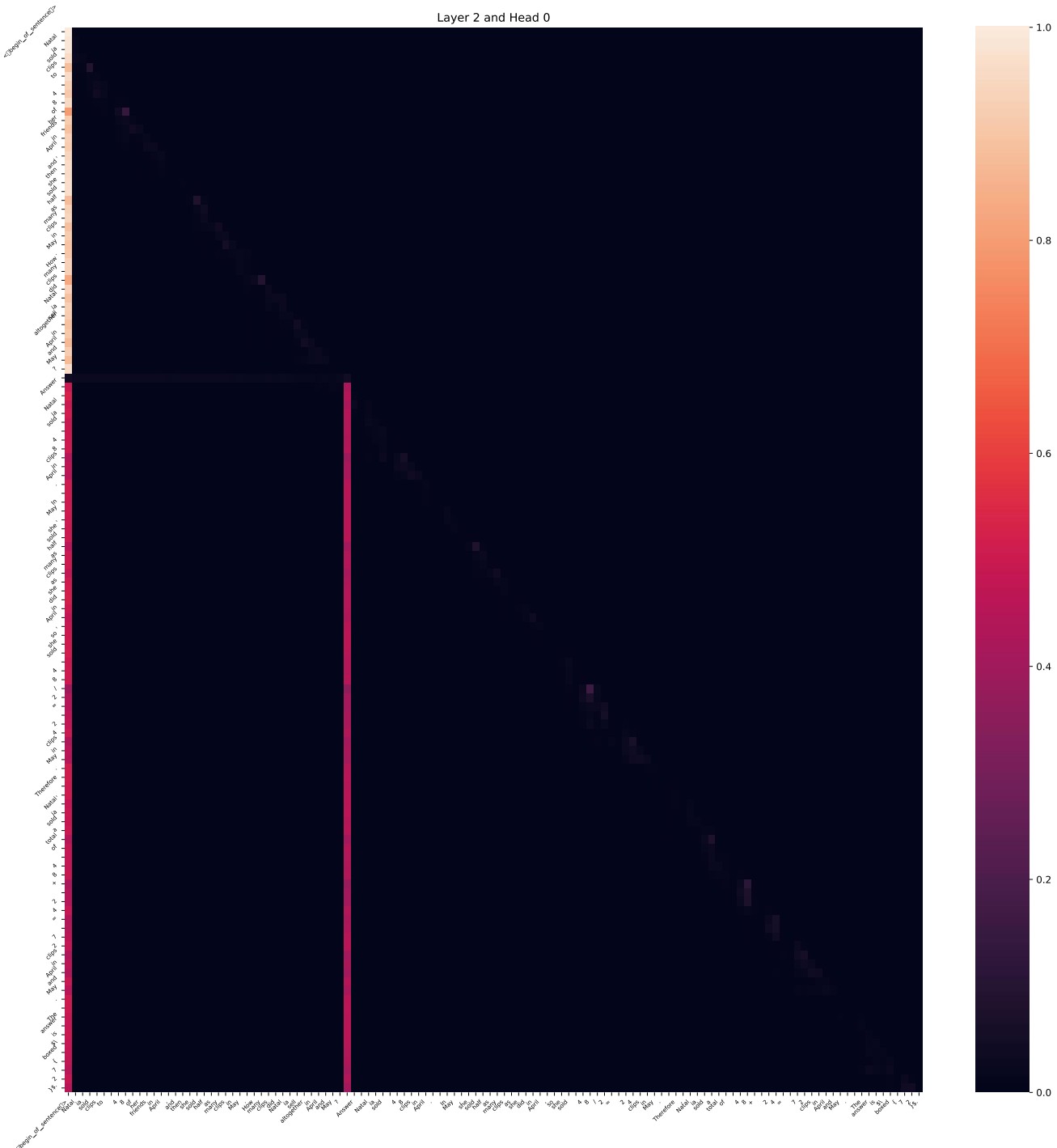

*Figure 14.* An example of attention map in Llama-3-8B-Instruct (Layer 2 and Head 0).

