# OpenReview forum: "SepLLM: Accelerate Large Language Models by Compressing One Segment into One Separator"
_ICML.cc/2025/Conference — ICML 2025 poster_

### Official Review · Reviewer_kB2E · 2025-03-10

**Overall Recommendation:** 3

**Summary:**

This paper proposes SepLLM, an efficient Transformer-based architecture designed to accelerate inference in large language models (LLMs). The key insight is that separator tokens (such as punctuation and line breaks) disproportionately carry segment-level information, enabling the compression of other tokens without significant performance loss. The authors implement a sparse attention mechanism that retains only initial, separator, and neighboring tokens, substantially reducing the memory usage of the KV cache. Experimental results demonstrate that SepLLM achieves more than 50% reduction in KV cache size on GSM8K-CoT while maintaining competitive accuracy, and significantly improves inference speed and memory efficiency in both static and streaming tasks.

**Claims And Evidence:**

The paper identifies separator tokens ([".", ",", "?", "!", ";", ":", " ", "\t", "\n"]) as critical for compressing segment-level information.
However, the authors do not clearly explain or experimentally justify how these separator tokens were chosen.

**Essential References Not Discussed:**

To the best of my knowledge, the paper has cited all essential references necessary to understand the context and key contributions of the work.

**Experimental Designs Or Analyses:**

The experimental design, particularly using GSM8K-CoT and MMLU, is well-structured to demonstrate efficiency and reasoning improvements. However, it remains unclear whether the approach can maintain performance on more challenging tasks requiring extended reasoning (e.g., Math 500), and further experiments in this area would be beneficial.

**Methods And Evaluation Criteria:**

The proposed methods, particularly the selective retention of initial, separator, and neighboring tokens, are well-aligned with the goal of reducing computational complexity in LLMs.
Additionally, employing well-established benchmarks such as GSM8K-CoT, MMLU, and PG19 provides a clear and appropriate evaluation framework for the model's performance.

**Other Comments Or Suggestions:**

In the Related Work, after discussing the shortcomings of existing KV cache compression methods, the paper does not explicitly highlight how its approach overcomes these limitations.

**Other Strengths And Weaknesses:**

Strengths:
1.The paper introduces an innovative approach by leveraging separator tokens to compress segment-level information, which is a novel idea in efficient LLM design.
2.The method is thoroughly evaluated across various settings (training-free, training-from-scratch, post-training, and streaming), demonstrating its practical effectiveness.
3.The approach achieves substantial computational savings—significantly reducing KV cache usage—while maintaining competitive performance.

Weaknesses:
1.The size of the neighboring tokens must be manually selected, requiring a trade-off between efficiency and performance.
2.The justification for the chosen set of separator tokens is not fully detailed, raising questions about the robustness of this selection across different tasks or languages.
3.The experiments are largely limited to benchmarks like GSM8K-CoT and MMLU, leaving uncertainty about the method’s effectiveness on more complex, reasoning-intensive tasks.

**Questions For Authors:**

1.How did you select the fixed set of separator tokens, and have you evaluated alternative sets?

2.Can you provide insights into how the size of neighboring tokens (n) might be determined adaptively?

3.How does SepLLM perform on more challenging, reasoning-intensive tasks beyond GSM8K-CoT and MMLU?

**Relation To Broader Scientific Literature:**

The paper builds on prior research in KV cache compression and sparse attention—such as FastGen, SnapKV, and StreamingLLM—by introducing a novel approach that leverages separator tokens to compress segment-level information. This contribution extends methods like BigBird and other sparse attention mechanisms by showing that dynamic, data-dependent token selection can significantly reduce computational overhead while maintaining performance.

**Theoretical Claims:**

I reviewed the theoretical proofs provided, particularly the universal approximation analysis in Appendix G.

---

> ### Author Rebuttal · Authors · 2025-04-01
>
> We would like to sincerely thank the reviewer for the valuable comments on our work. We take every comment seriously and hope our response can address the reviewer’s concerns. If there are any remaining questions, we are more than happy to address them.
>
> > Q1. The authors do not clearly explain or experimentally justify how these separator tokens were chosen.
>
> **A1**.
>  We take the tokens in the 300B natural language dataset that exceed a certain word frequency threshold as separators, which are [“.”, “,”, “?”, “!”, “;”, “:”, “ ”, “\t”, “\n”].
>
> > Q2. The size of the neighboring tokens must be manually selected, requiring a trade-off between efficiency and performance.
>
> **A2**.
> In our experiments, the size of neighboring tokens is set to be $m/8$ ~ $m/4$, where $m$ denotes the input sequence length. We find that the performance is still closed to that of full attention in these settings. While the size of neighboring tokens is set to be $m/2$ in StreamingLLM to maintain a similar performance.
>
> > Q3. The justification for the chosen set of separator tokens is not fully detailed, raising questions about the robustness of this selection across different tasks or languages.
>
> **A3**.
> Indeed, we use the same set of separators in all the experiments, including using lama-3-8B, Pythia-6.9B, Falcon-40B models and conducted on GSM8K, ARC, and MMLU datasets.
>
> > Q4. The experiments are largely limited to benchmarks like GSM8K-CoT and MMLU, leaving uncertainty about the method’s effectiveness on more complex, reasoning-intensive tasks.
>
> **A4**.
>
> As suggested, we evaluate our SepLLM on MATH, which is a more complex benchmark, and the results are as follows.
> It justifies the superiority of SepLLM in complex tasks.
>
> | | MATH | | | | | | | | |
> |---|---|---|---|---|---|---|---|---|---|
> | |**overall** | **r.KV(%)** |algebra | counting_and_prob | geometry |intermediate_algebra | num_theory | prealgebra | precalc |
> | Vanilla |27.32| 100.00|37.83| 24.26|18.58|12.62|21.11| 48.34|11.72|
> | SepLLM | 26.74 |32.4 | 37.99 | 24.05 | 19.83 |10.52|18.7|48.56|10.62 |
> | H$_2$O | 23.72 | 33.1| 34.71 | 17.72 | 15.03 | 11.41 | 19.63 | 41.79 | 8.24|
> | SnapKV | 25.68 | 33.1 | 37.49 | 21.94 | 16.91 | 10.41 | 20.37 | 45.58 |9.71|
> | PyramidKV | 25.1 | 33.1| 37.49|20.46| 16.49 | 9.97 |19.44 |44.55 |9.34|
>
>
> >   Q5. In the Related Work, after discussing the shortcomings of existing KV cache compression methods, The paper does not explicitly highlight how its approach overcomes these limitations.
>
> **A5**.
> As discussed in Section 1 of our submitted paper, the existing KV cache compression methods are training-free approaches, which is inconsistent with training phase, resulting in approximation error.
>
> > Q6. How did you select the fixed set of separator tokens, and have you evaluated alternative sets?
>
> **A6**.
> For the selection confusion, please refer to Q1.
> As suggested, we evaluate our SepLLM with subsets of our previous setting on GSM8K-CoT, which are  ["," "."  "?"  ";"] and ["."  "?"]. The experimental results are as follows. It shows that performance drops <0.5 when removing half of the separators. So SepLLM is robust against the separator selection.
>
> |                  | SepLLM (n=256)   | SepLLM (n=256)                | SepLLM (n=256) | StrmLLM (n=256) | StrmLLM (n=380) |
> |------------------|-------------------------------|----------------|-----------------|-----------------|-----------------|
> | flexible-extract | 77.18                         | 76.68          | 70.66           | 68.84           | 71.42 |
> | strict-match     | 77.18                         | 76.85          | 70.36           | 67.63           | 70.89 |
> | r.KV (%)         | 47.36                         | 37.92          | 36.44           | 31.92           | 47.54 |
> | separators       | "." "," "?" "!" ":" ":" " " "\t" "\n"  | "," "."  "?"  ";" | "."  "?"          | - | - |
>
> > Q7. Can you provide insights into how the size of neighboring tokens (n) might be determined adaptively?
>
> **A7**.
> As discussed in A2, we recommend the size of neighboring tokens to be $m/8$~$m/8$, where $m$ denotes the input sequence length.
>
> Q8. How does SepLLM perform on more challenging, reasoning-intensive tasks beyond GSM8K-CoT and MMLU?
>
> **A8**.
> Please refer to Q4.

---

### Official Review · Reviewer_Crr7 · 2025-03-14

**Overall Recommendation:** 3

**Summary:**

## update after rebuttal
The author's rebuttal addressed many of my concerns; however, I am still a little hesitant about the generalisability. I will change my score from 2 to 3.


* The paper identifies a key pattern: certain seemingly meaningless special tokens (i.e., separators) contribute disproportionately to attention scores compared to semantically meaningful tokens.
* Plug-and-play method for compressing such special tokens and removing redundancy.
* Also implement efficient kernels for training acceleration.

**Claims And Evidence:**

* "From Figure 2,  it is seen that a given input context, seemingly “meaningless” separator tokens (such as commas, periods, exclamation marks, semicolons, etc.) that segment sequences receive higher attention scores compared to semantically meaningful tokens
(such as nouns or verbs)." This is a very interesting observation and is well used for the design.

* "Neighboring tokens usually help form locally smooth and coherent contexts, allowing the model to generate sentences that are reasonable within the immediate context." I like how the authors well defined all the different kinds of tokens.

* Targeted masking for pre-training so that different kinds of tokens have different importance score follows well from the initial insights.

**Essential References Not Discussed:**

* Comparison to popular KVCache compression methods such as H20 (Neurips '23) and SnapKV (Arxiv'24) has not been considered. Would love to see insights on that too.

**Experimental Designs Or Analyses:**

* The experimental setup was well explained and evaluation on many parameters (quality, flops (compute) and size ) was carried out.
* It was useful to see both the pre-training and post-training performance
* One issue in the ablation studies, the claim of better retrieval was introduced but no result was shown in the main body of the paper.

**Methods And Evaluation Criteria:**

* Splitting the KV Cache into different blocks based on the kind of tokens seems like a promising idea and is well explained.
* Evaluation is carried out on standard benchmark datasets and show very promising results
* "Our positional encoding strategy for streaming settings is the same as the state-of-the-art StreamingLLM" more information on the exact technique would have been helpful, as the correct positional encoding seems to inspire the design a lot. Is the technique used the same as ROPE? Better?
* Ablation studies show the importance of additional tokens, but it seems a better idea would have been to show the effect of separator tokens as that is the main claim for improvement.
* How does SepLLM compare against well known KVCache compression techniques such as H20 and SnapKV?

**Other Comments Or Suggestions:**

N/A

**Other Strengths And Weaknesses:**

* Interesting approach to see a KVCache compression technique which considers the KVCache being split into multiple sub KVCaches based on the properties of the tokens.

**Questions For Authors:**

I'd encourage the authors to compare with well known KVCache compression techniques such  H20 (Neurips '23) and SnapKV (Arxiv'24) to better position the gains and have further clarity on how much the token-based KVCache partitioning helps. I saw a claim in the introduction that training-free methods are suboptimal but no evidence has been presented in the paper.

**Relation To Broader Scientific Literature:**

* The space of KV Cache compression is extremely interesting and has very promising implications on faster inference for LLMs. The scope and the methods in the paper are timely and well presented.

**Theoretical Claims:**

Minor theoretical results are presented and the claims seem correct.

---

> ### Author Rebuttal · Authors · 2025-04-01
>
> We would like to sincerely thank the reviewer for the valuable comments on our work. We take every comment seriously and hope our response can address the reviewer’s concerns. If there are any remaining questions, we are more than happy to address them.
>
> **Should there be any need for further clarification to assist in advancing our scores, please do not hesitate to inform us. Thank you very much.**
>
> >  Q1. "Our positional encoding strategy for streaming settings is the same as the state-of-the-art StreamingLLM" more information on the exact technique would have been helpful, as the correct positional encoding seems to inspire the design a lot. Is the technique used the same as ROPE? Better?
>
> **A1**.
> We just adopt RoPE. However, in scenarios involving infinitely long streaming input, we follow StreamingLLM[1] and apply PE's shifting; i.e., we focus on positions within the cache instead of those in the original text. For instance, if the current cache contains tokens [0, 1, 2, 3, 6, 7, 8] and is in the process of decoding the 9th token, the positions assigned are [0, 1, 2, 3, 4, 5, 6, 7], rather than the positions in the original text, which would be [0, 1, 2, 3, 6, 7, 8, 9]. However, for training and downstream tasks of general length (<4K; Sec 3.1), we simply use standard RoPE without PE's shifting. See **Sec. 3.3;4.6 and Tab. 8** for details.
>
> > Q2. Ablation studies show the importance of additional tokens, but it seems a better idea would have been to show the effect of separator tokens as that is the main claim for improvement.
>
> **A2**.
> Indeed, the ablation studies you propose are illustrated in **Tab.1-8 and Fig. 5** (for both training & training-free). Without separators, SepLLM degrades to StreamingLLM. As demonstrated by the experimental results across so many settings and tasks, the separators indeed provide significant benefits to the architecture design. See more studies on separators in **A6 to Reviewer kB2E**.
>
> > Q3. How does SepLLM compare against well known KVCache compression techniques such as H2O and SnapKV?
>
> **A3**.
> As suggested, we add the H2O, SnapKV and PyramidKV [2] as baselines and the results on GSM8K_CoT and the challenging MATH are as follows (All based on Llama3-8B-Inst backbones).
> | | GSM8K-CoT | | |
> |---|---|---|---|
> | | flexible | strict | r.KV(%) |
> | Vanilla |77.79| 77.26| 100.00|
> | SepLLM | 77.18 |77.18| 47.36|
> | H2O | 76.27 |75.06| 47.54|
> | SnapKV | 76.5 |73.62| 47.54|
> | PyramidKV | 75.82 |72.02| 47.54|
>
> | | MATH | | | | | | | | |
> |---|---|---|---|---|---|---|---|---|---|
> | |**overall** | **r.KV(%)** |algebra | counting_and_prob | geometry |intermediate_algebra | num_theory | prealgebra | precalc |
> | Vanilla |27.32| 100.00|37.83| 24.26|18.58|12.62|21.11| 48.34|11.72|
> | SepLLM | 26.74 |32.4 | 37.99 | 24.05 | 19.83 |10.52|18.7|48.56|10.62 |
> | H$_2$O | 23.72 | 33.1| 34.71 | 17.72 | 15.03 | 11.41 | 19.63 | 41.79 | 8.24|
> | SnapKV | 25.68 | 33.1 | 37.49 | 21.94 | 16.91 | 10.41 | 20.37 | 45.58 |9.71|
> | PyramidKV | 25.1 | 33.1| 37.49|20.46| 16.49 | 9.97 |19.44 |44.55 |9.34|
>
> With similar amount of KV cache, SepLLM can achieve better performance.
>
> H2O and SnapKV are training-free methods that rely on full attention for pretraining and prefilled KV, but their inference-stage KV selection causes training-inference inconsistency and query dependency. SepLLM addresses these issues by introducing **a novel language modeling paradigm** that uses separators for segment infomation summarization, **optimizing from the pretraining to the inference**. Additionally, SepLLM naturally aligns with the hierarchical semantic structure of language (e.g., "," for segments, "." for sentences, \n for paragraphs) and reduces KV usage during inference.
>
> > Q4. One issue in the ablation studies, the claim of better retrieval was introduced but no result was shown in the main body of the paper.
>
> **A4**.
> Indeed, we have already conducted **needle in a haystack** experiments and the results are shown in **Fig. 8-11** &  App. E, which demonstrate the retrieval ability of SepLLM.
>
> > Q5. Comparison to popular KVCache compression methods such as H20 (Neurips '23) and SnapKV (Arxiv'24)
>
> **A5**.
> Please refer to Q3.
>
> > Q6. I saw a claim in the introduction that training-free methods are suboptimal but no evidence has been presented in the paper.
>
> **A6**.
> As discussed in **Q3**, training-free methods are based on KV selection, an approximation to full attention, leading to
> inconsistency between training and inference phases.
>
> **Tab.13** can support our claim. Based on the well-trained Llama3-8B-Inst model, we fine-tuned for only 200 steps on the LongAlpaca dataset, and the results outperformed the original training-free SepLLM and even surpassed Vanilla.
>
> [1] Xiao, Guangxuan, et al. "Efficient streaming language models with attention sinks." ICLR 2024.
> [2] Cai, Zefan, et al. "Pyramidkv: Dynamic kv cache compression based on pyramidal information funneling." preprint arXiv:2406.02069 (2024).

---

### Official Review · Reviewer_pjB2 · 2025-03-15

**Overall Recommendation:** 3

**Summary:**

The paper introduces SepLLM, a novel framework aimed at improving the efficiency of large language models (LLMs) by leveraging the observation that separator tokens (e.g., commas, periods) disproportionately contribute to attention scores. The authors propose compressing segment information into these separator tokens, reducing computational costs and memory usage while maintaining model performance.

## update after rebuttal

The author's rebuttal addressed most of my concerns; I will keep my score.

**Claims And Evidence:**

Yes, the authors propose a sparse attention mechanism based on separator tokens and validate its effectiveness through experiments.

**Essential References Not Discussed:**

None

**Experimental Designs Or Analyses:**

- Although there are relatively sufficient experiments on long texts, some important evaluations are still missing, such as long ppl [1], because traditional ppl evaluations may not accurately reflect long-text capabilities on truly extended texts. Additionally, testing tasks like "needle in a haystack" would help assess long-text capabilities.
- Furthermore, some simple baselines were not considered. For instance, while this method retains separator tokens, what would happen if we retained one token for every fixed number of tokens? Especially after some training, fixed sparse strategies might also prove to be quite efficient.

[1] What is Wrong with Perplexity for Long-context Language Modeling?

**Methods And Evaluation Criteria:**

Yes, the authors first observe the attention matrix and find that separator tokens have stronger attention weights, which leads to the proposal of the SepLLM method. And the evaluation on long-context ppl is the standard setting in sparse attention.

**Other Comments Or Suggestions:**

None

**Other Strengths And Weaknesses:**

The reliance on separator-based design might affect generalization, as separators may vary across different languages or datasets. A more general consideration could involve first performing adaptive chunking on the data, ensuring low entropy within chunks and high entropy between chunks, followed by compressing each chunk.

**Questions For Authors:**

- What would happen if sparse attention is not used, and instead some heads are converted to sliding window attention while allocating the remaining computation to a full window attention? This would represent a standard hybrid baseline.
- Are there any methods to further improve the KV compression rate of SepLLM?

**Relation To Broader Scientific Literature:**

The key contributions of the paper are closely related to prior research on sparse attention mechanisms and efficient memory management in large language models (LLMs). The observation that separator tokens receive disproportionately high attention aligns with earlier findings on the importance of specific tokens (e.g., attention sinks) in long-context modeling, as highlighted in works like StreamingLLM and SnapKV. Additionally, the proposed SepLLM framework builds on ideas from sparse attention methods, such as BigBird and Longformer, but differentiates itself by focusing on data-dependent token selection (separator tokens) rather than fixed patterns.

**Theoretical Claims:**

N/A

---

> ### Author Rebuttal · Authors · 2025-04-01
>
> We would like to sincerely thank the reviewer for the valuable comments on our work. We take every comment seriously and hope our response can address the reviewer’s concerns. If there are any remaining questions, we are more than happy to address them.
>
> **Should there be any need for further clarification to assist in advancing our scores, please do not hesitate to inform us. Thank you very much.**
>
> > Q1. important evaluations such as long ppl and "needle in a haystack"
>
> **A1**.
> We have provided numerous results on perplexity (ppl) for extremely long test datasets. For instance, based on the Llama3-8B model and the PG19 test set, we conducted inference on extremely long sequences of up to 4 million tokens. The results are presented in Table 4 of our paper, which we have copied here for your reference.
>
> | Input Length |3M |3.5M | 4M |
> |----|----|----|----|
> |StreamingLLM| 36.4|35.8|36.1|
> | SepLLM(s=32)|34.9|34.2|34.5|
> |SepLLM(s=64)|34.3|33.7|33.9|
>
> Additionally, we have also conducted experiments on **Needle In A Haystack**, with results and detailed discussions presented in Figure 8-11 of our submitted paper. The experimental results show the long-context retrivie capacity of SepLLM.
>
> > Q2. retained one token for every fixed number of tokens
>
> **A2**.
> As suggested, we conducted such experiments and the results are shown in the table below.
>
> | | GSM8K-CoT | | | MMLU |  |  |  | |   |
> |---|---|---|---|---|---|---|---|---|---|
> | | flexible | strict | r.KV(%) | humanities | stem | social | other | Overall |  r.KV(%) |
> |Vanilla |77.79|77.26|100.00|60.49|56.61|76.50|72.19|65.72|100.00|
> | FixLLM($\Delta$=5,$n$=256)|70.43|70.43|45.64|55.52|54.80|72.99|69.75|62.33|50.20|
> | FixLLM($\Delta$=4,$n$=256)|72.71|72.33|49.08|55.92|54.39|74.36|70.81|62.91|53.32|
> |SepLLM($n$=256)| 77.18| 77.18| 47.36| 57.66| 56.49| 76.21| 72.19| 64.68| 44.61|
>
> Here, FixLLM represents the variant you mentioned, where $n$ still denotes the number of neighboring tokens, and every $\Delta$-th tokens are also remained outside the neighboring window. Experimental results indicate that FixLLM is still inferior to SepLLM.
>
>
> > Q3.  generalization issue as separators may vary across different languages or datasets.
>
> **A3**.
> In our perspectives, separators are general across different languages or datasets. In modern commonly used languages, separators are widely present (including but not limited to English, German, French, Italian, Chinese, and etc, covering over 7 Billion people). While most languages use separators similar to modern English, a few languages still retain the usage of their traditional separators. For example, in Bengali, the period (".") is commonly used as a sentence-ending marker, but the traditional sentence-ending marker "।" (vertical line) is still in use. During training and inference, we only need to specify the separators we want to use, and the commonly used separators in most languages are around 10 in number.
>
> Emprically, we have verified the robustness of our SepLLM across various tasks or datasets, including GSM8K, MMLU, ARC, PIQA and so on. All emprical results can demonstrate the generalization of SepLLM as well.
>
> > Q4. low entropy within chunks and high entropy between chunks
>
> **A4**.
> We agree that entropy-based grouping is a feasiable apporach. However, entropy-based grouping is not training/inference efficient since we need to calculate the attention mask by some algorithms. In comparison, SepLLM follows the natural modeling of the language and split the segments by separators, which is more efficient and more robust.
>
> > Q5.  some heads are converted to sliding window attention while allocating the remaining computation to a full window attention? This would represent a standard hybrid baseline.
>
> **A5**.
> As suggested, we added the following experiments.
>
> | | MMLU | | | | |r.KV(%) | n|
> |---|---|---|---|---|---|---|---|
> | #Sparse_Heads/#Total_Heads | humanities | social sciences | stem | other |overall |	| |
> |20/32 | 23.93 | 23.07 | 24.42 | 23.53 | 23.76|44.74|80|
> |24/32 | 24.23 | 23.3 | 26.36 | 23.37 | 24.31|47.71|208|
> |28/32 | 25.66 | 27.29 | 26.31 | 23.69 | 25.81|45.13|256|
> |30/32 | 27.29 | 25.09 | 27.78 | 38.11 | 29.31|45.42|288|
> |SepLLM	|57.66|76.21| 56.49| 72.19|64.68|44.61|256|
> |Vanilla | 60.49| 76.50 |  56.61 | 72.19 | 65.72|100||
>
> For example, "20/32" means there are 20 sliding-window heads out of a total of 32 heads (the remaining heads using full attention).
> As can be seen, with similar amount of KV cache, SepLLM is much better than these hybrid baselines.
>
> > Q6. any methods to further improve the KV compression rate of SepLLM
>
> **A6**.
> One possible method may be the multi-level version of SepLLM. In the current version, we compress the "segment-level" into general seperators. In the multi-level version, we may define "sentence-level" or "paragraph-level" as a higher-level seperation and compress a set of previous seperators into one more condensed KV Cache. Then, the KV compression rate will be improved.

---

### Decision · Program_Chairs · 2025-05-01

**Decision:**

Accept (poster)

**Comment:**

This paper presents a novel attention mechanism that allows LLMs to attend to only a subset of previous tokens (a few initial tokens, separator tokens, and local neighboring tokens). This approach improves both training and inference speed, and reduces the size of the KV cache necessary at inference time. Reviewers agree the paper is well motivated, the method is well explained, and that the evaluations are thorough and show the proposed approach is strong. The initial reviews raised valid concerns, including comparisons to additional baselines (like uniformly subsampling tokens to attend to), and robustness of the design choices (e.g. the separator tokens). The authors have addressed most of these in the rebuttal, and reviewers have appropriately raised their scores. Given the performance of the method and that reviewers agree on acceptance, we recommend this paper for acceptance at ICML.